# Does higher interpretability imply better utility? A Pairwise Analysis on Sparse Autoencoders

**Xu Wang**[1,2,3*]**, Yan Hu**[2,4]**, Benyou Wang**[2,3]**, Difan Zou**[1,3†]
[1]The University of Hong Kong
[2]The Chinese University of Hong Kong, Shenzhen
[3]Shenzhen Loop Area Institute
[4]National Health Data Institute, Shenzhen

## Abstract

Sparse Autoencoders (SAEs) are widely used to steer large language models (LLMs), based on the assumption that their interpretable features naturally enable effective model behavior steering. Yet, a fundamental question remains unanswered: *does higher interpretability indeed imply better steering utility?* To answer this question, we train 90 SAEs across three LLMs (Gemma-2-2B, Qwen-2.5-3B, Gemma-2-9B), spanning five architectures and six sparsity levels, and evaluate their interpretability and steering utility based on SAEBench (Karvonen et al., 2025) and AxBench (Wu et al., 2025) respectively, and perform a rank-agreement analysis via Kendall's rank coefficients $\tau_b$. Based on the framework, our analysis reveals only a relatively weak positive association ($\tau_b \approx 0.298$), indicating that interpretability is an insufficient proxy for steering performance. We conjecture the interpretability-utility gap may stem from the selection of SAE features as not all of them are equally effective for steering. To further find features that truly steer the behavior of LLMs, we propose a novel selection criterion: $\Delta$ *Token Confidence*, which measures how much amplifying a feature changes the next token distribution. We show that our method improves the steering performance of three LLMs by **52.52%** compared to the current best output score-based criterion (Arad et al., 2025). Strikingly, after selecting features with high $\Delta$ *Token Confidence*, the correlation between interpretability and utility vanishes ($\tau_b \approx 0$), and can even become negative. This further highlights the divergence between interpretability and utility for the most effective steering features. Code is available at `https://github.com/Xu0615/SAE4Steer`.

## 1 Introduction

As Large Language Models (LLMs) become more widely used in real-world applications, ensuring the safety of their outputs is increasingly important (Kumar et al., 2024; Ji et al., 2023; Inan et al., 2023). Reliable and controllable behavior is essential for deploying these LLMs in more situations (Chen et al., 2024). Fine-tuning is the standard way to improve controllability, but it requires labeled data, significant training time, and compute resources (Hu et al., 2022; Wang et al., 2025c). This has spawned a series of representation-based interventions, i.e., steering, that guide LLM inference by manipulating internal representations, aiming for faster and more lightweight output control (Turner et al., 2023; Turner et al., 2024; Wang et al., 2025b; Stolfo et al., 2025).

However, activation-level edits are often coarse: they mix multiple semantics, a phenomenon called polysemanticity (Bricken et al., 2023). Recently, Sparse Autoencoders (SAEs) have become a valuable tool in the interpretability field. They are trained to actively decompose the hidden states of the LLM into sparse and human readable features (Templeton et al., 2024; Mudide et al., 2025).

---

*First author. Email: `sunny615@connect.hku.hk`
†Corresponding author. Email: `dzou@hku.hk`

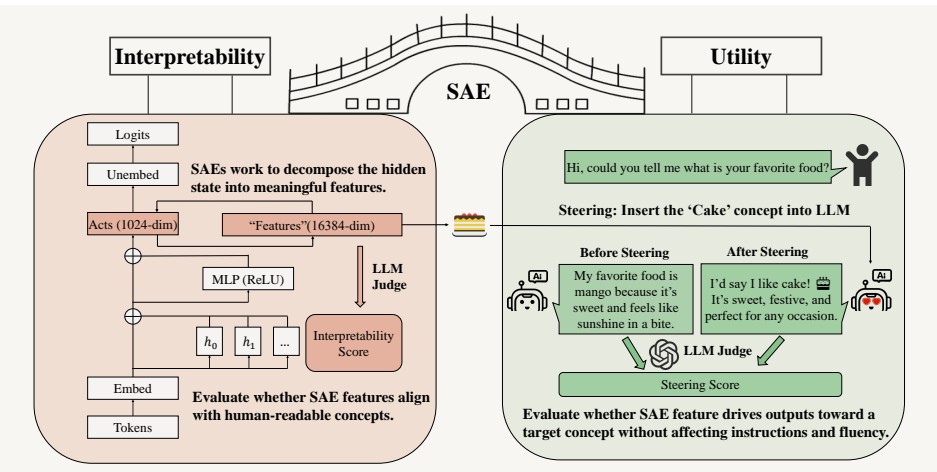

Figure 1: **Overview of our goal: building a bridge for SAE interpretability and utility. Interpretability (left):** an SAE attached to the LLM decomposes hidden states into sparse, human-describable features. An LLM judge yields an *interpretability score* for the SAE (Paulo et al., 2025). **Utility (right):** at inference, we activate a target SAE feature (e.g., 'cake') to steer generation. An LLM judge yields *steering utility score* (Wu et al., 2025).

Their interpretable nature has subsequently spurred research into leveraging SAE features for more precise, concept-level control over model behavior (Ferrando et al., 2025; Chalnev et al., 2024).

Despite this progress, a critical question remains unanswered: *does higher interpretability truly imply better utility?* Since SAEs are trained to balance reconstruction and sparsity to yield human-readable features (Cunningham et al., 2023; Makelov, 2024; O'Brien et al., 2025), their utility for downstream tasks is not a primary objective. Understanding and characterizing this gap is critical to enabling more interpretable and effective steering with SAEs, because (i) it clarifies when interpretability scores can (and cannot) be used as a practical proxy for utility, and (ii) it motivates training objectives that explicitly balance reconstruction, interpretability, and downstream steering performance. To this end, we conduct a systematic study to build a bridge between SAE interpretability and steering utility (see Figure 1).

To perform a comprehensive association analysis, we train 90 SAEs across three LLMs (Gemma-2-2B (Team et al., 2024), Qwen-2.5-3B (Yang et al., 2024), and Gemma-2-9B) spanning diverse architectures and sparsity levels. We compute interpretability using SAEBENCH (Karvonen et al., 2025) and steering utility using AXBENCH (Wu et al., 2025). Then, we leverage a pairwise-controlled framework to evaluate whether interpretability predicts steering performance across the pool of trained SAEs. To quantify this relationship, we follow the idea of prior works (Jiang et al., 2020; Hu et al., 2024) and measure rank agreement between interpretability and utility using Kendall's rank coefficient $\tau_b$. We control confounders with an axis-conditioned analysis, isolating each design axis (architecture, sparsity, model) by varying one at a time and aggregating per-axis metrics.

Furthermore, as identified in Arad et al. (2025); Wu et al. (2025), not all interpretable features in SAE are equally effective for steering. This motivates our next objective to identify the specific features critical for behavior control and steering utility analysis. Motivated by the recent progress on the entropy mechanism in LLM reasoning (Fu et al., 2025; Wang et al., 2025a), we propose an innovative selection criterion for SAE features: $\Delta$ *Token Confidence*, which measures the degree to which amplifying a single feature shifts the model's next-token distribution. Features that induce the most substantial change in model confidence are identified as high-utility candidates features for steering, as they exert a measurable and targeted influence on model behavior. Finally, we leverage these critical features to conduct a refined analysis of the interpretability-utility gap.

The primary contributions and insights of this paper are summarized as follows:

1. **(§3.4) Interpretability shows a relatively weak positive association with utility.** Across 90 SAEs that are trained across three model sizes, five architectures, and six sparsity levels, we

find that a higher *interpretability score* tends to shows a relatively weak positive association with steering performance (the Kendall's rank coefficient $\tau_b \approx 0.298$)). This identifies a notable interpretability-utility gap of the existing SAEs.

2. **(§4.2) $\Delta$ *Token Confidence* effectively selects features with strong steering performance.** To identify the SAE features that are critical for steering, we introduce $\Delta$ *Token Confidence*, an innovative metric that identifies steering-critical SAE features by measuring their impact on the model's next-token distribution. When benchmarked against the best existing output score-based method (Arad et al., 2025), our approach yields a substantial 52.52% average improvement in *steering score*. This result validates the superiority of our method and underscores the critical role of feature selection in characterizing and enhancing the steering utility of SAEs.

3. **(§4.3) The interpretability-utility gap widens among high-utility features.** By reapplying our association analysis exclusively to SAE features with strong steering utility, we uncover a counterintuitive finding: the interpretability-utility correlation vanishes or even becomes negative (Kendall's rank coefficient $\tau_b \approx 0$). This indicates that for the most effective steering features, interpretability is at best irrelevant and potentially detrimental, further emphasizing the critical nature of the interpretability-utility gap.

Our results demonstrate a significant divergence between SAE's interpretability and steering utility, suggesting that prioritizing interpretability does not enable improved steering performance. This gap highlights a crucial research direction: mitigating it will likely necessitate advanced post-training feature selection protocols or fundamentally new, utility-oriented SAE training paradigms.

## 2 PRELIMINARY

### 2.1 SPARSE AUTOENCODERS

Sparse Autoencoders (SAEs) decompose internal model activations $x$ into sparse, higher-dimensional features $h$ that can be linearly decoded back to the original space (Cunningham et al., 2023; Leask et al., 2025). A standard SAE with column-normalized decoder weights (Bricken et al., 2023; Karvonen et al., 2024) is defined by the following forward map and optimization objective:

$$\mathcal{L} = \|x - \hat{x}\|_2^2 + \lambda \|h\|_1, \text{ where } h = \text{ReLU}(W_E x + b_E), \hat{x} = W_D h + b_D,$$

where $W_E, b_E$ are encoder parameters, $W_D, b_D$ are decoder parameters, $\hat{x}$ is the reconstruction, and $\lambda$ controls sparsity. This training balances reconstruction accuracy with sparse representations.

### 2.2 INTERPRETABILITY: AUTOMATED INTERPRETABILITY SCORE

SAEBENCH (Karvonen et al., 2025) uses an LLM-as-judge (Paulo et al., 2025) to assess each latent: the judge drafts the description from examples and then predicts, on a held-out set, which sequences activate it. The *Automated Interpretability Score* is the average precision of the judge's prediction.

$$\text{AutoInterp Score} = \frac{1}{M} \sum_{m=1}^{M} \mathbf{1}[\hat{y}_m = y_m],$$

where $y_m \in \{0, 1\}$ indicates whether the latent activates in the sequence $m$ and $\hat{y}_m$ is the judge's prediction. We use this score as our interpretability metric. For the complete details, see Appendix B. Throughout this paper, *interpretability* refers specifically to the interpretability of SAE features as measured by automated metrics (e.g., SAEBENCH), rather than interpretability methods in general.

### 2.3 UTILITY: STEERING SCORE

SAE steering injects the SAE decoder atom $v_f$ (the $f$-th column of the column-normalized decoder $W_{\text{dec}}[f]$) into the residual stream at a target layer to push the hidden state $x$ along a chosen feature direction (Durmus et al., 2024). Given a feature index $f$, a steering factor $\alpha$, and a per-sample scale $m_f$ (e.g., the feature's maximum activation), the intervention is

$$x^{\text{steer}} = x + (\alpha m_f) \cdot v_f. \tag{1}$$

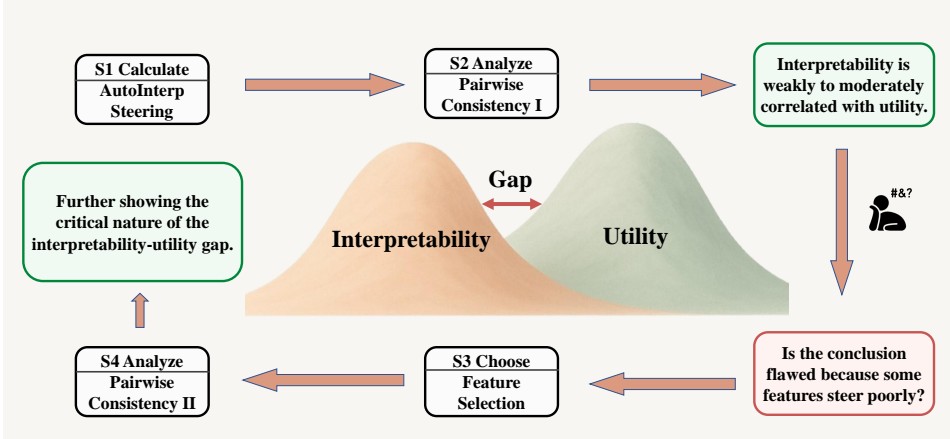

Figure 2: **Overview of our pairwise-controlled workflow linking SAE interpretability with steering utility.** **(S1)** Compute *interpretability score* and *steering score* for each SAE. **(S2)** Pairwise analysis across SAEs and get an insight (the top-right green box), revealing an interpretability–utility gap. The red box (lower right) is our further inference based on the above green box and previous studies (Wu et al., 2025). **(S3)** Use $\Delta$ *Token Confidence* to select higher-utility features. **(S4)** Compute *steering score* after selection per SAE, then do the pairwise analysis between *steering score* and *interpretability*. The green box in the middle left is our final conclusion.

Through the above formula (1), we can use SAE features for steering to achieve the output of controlling LLM. AXBENCH (Wu et al., 2025) measures causal control by steering internal representations during generation and asking an LLM judge to rate three aspects, each on a discrete scale $\{0, 1, 2\}$: *Concept* ($C$), *Instruction* ($I$), and *Fluency* ($F$). The overall *Steering Score* is the harmonic mean:

$$\text{Steering Score} \;=\; \text{HM}(C, I, F) \;=\; \frac{3}{\frac{1}{C} + \frac{1}{I} + \frac{1}{F}} \;\in\; [0, 2].$$

Following AXBENCH, for each concept we sample instructions (e.g., 10 from Alpaca-Eval (Dubois et al., 2023)), generate continuations under different steering factors, pick the best factor on one split, and evaluate the held-out split with the judge to obtain the final utility score averaged across prompts (Gu et al., 2025). The complete scoring procedure is detailed in Appendix C. Likewise, our notion of *utility* is limited to *steering utility*—the effectiveness of using SAE features to causally steer model generations under the AXBENCH protocol.

## 3 CAN SAE INTERPRETABILITY INDICATE STEERING PERFORMANCE?

### 3.1 EXPERIMENTAL SETUP

**Dataset.** For each trained SAE, we score 1,000 latents with *LLM-as-judge* (Paulo et al., 2025) and randomly sample 100 to form that SAE's CONCEPT100: 100 human–readable concept descriptions per evaluation set, each pairing a latent (layer, feature_id) with a description (see details in Appendix F). For steering, we sample 10 Alpaca-Eval instructions, allow up to 128 generated tokens, and test 6 steering factors; the 10 instructions are split 5/5 for factor selection vs. held-out evaluation.

**Model.** We evaluate three open LLMs: Gemma-2-2B (Team et al., 2024), Qwen-2.5-3B (Yang et al., 2024), and Gemma-2-9B (Team et al., 2024). SAEs are trained on residual-stream activations at a fixed mid-layer for each model: Layer 12 for Gemma-2-2B, Layer 17 for Qwen-2.5-3B, and Layer 20 for Gemma-2-9B—and steering is applied to the corresponding layer.

**SAE with different architectures** We train 90 SAEs covering a range of architectures and sparsity. All SAEs use a latent dictionary width of 16k. We instantiate five variants: BatchTopK (Bussmann et al., 2024), Gated (Rajamanoharan et al., 2024a), JumpReLU (Rajamanoharan et al., 2024b), ReLU (Team, 2024), TopK (Gao et al., 2024) and sweep six target sparsity levels with approximate per-token activations $L_0 \approx 50, 80, 160, 320, 520, 820$. Further details are provided in Appendix A.

## 3.2 PAIRWISE RANK CONSISTENCY BETWEEN INTERPRETABILITY AND UTILITY

We test whether higher interpretability of SAE is predictive of higher steering performance across a set of trained SAEs attached to a fixed LM. For each SAE $\theta$ in a pool $\Theta$, we record a pair $(\mu(\theta), g(\theta)) \in \mathbb{R}^2$, where $\mu$ is the SAE-level *Interpretability Score* and $g$ is an aggregated *Steering Score* over a standardized evaluation suite.

Given two SAEs $\theta_i, \theta_j \in \Theta$, define the concordance indicator

$$v_{ij} = \text{sign}\big(\mu(\theta_i) - \mu(\theta_j)\big) \cdot \text{sign}\big(g(\theta_i) - g(\theta_j)\big) \in \{-1, 0, +1\}. \tag{2}$$

Kendall's tie-corrected rank coefficient $\tau_b$ (Kendall, 1938) summarizes agreement over unordered pairs and reduces to average concordance when there are no ties:

$$\tau_b = \frac{\displaystyle\sum_{i<j} v_{ij}}{\sqrt{\left(\displaystyle\sum_{i<j} \mathbf{1}\big[\mu(\theta_i) \neq \mu(\theta_j)\big]\right)\left(\displaystyle\sum_{i<j} \mathbf{1}\big[g(\theta_i) \neq g(\theta_j)\big]\right)}} \in [-1, 1], \tag{3}$$

In this study, we instantiate $\mu$ with the *Interpretability Score* and $g$ with the *Steering Score*, then compute $\tau$ for three model–layer settings (Gemma-2-2B, Qwen-2.5-3B, Gemma-2-9B). Each setting includes 30 SAEs spanning architectures and sparsity to ensure sufficient pair coverage.

## 3.3 GRANULATED KENDALL'S COEFFICIENT TO CONTROL CONFOUNDERS

Global rank agreement can be confounded by hyperparameters that jointly influence interpretability and utility. To obtain an axis-controlled assessment, we factor the SAE design space into orthogonal axes and evaluate rank consistency while varying one axis at a time and holding the others fixed.

We define three conditioning axes: (A) Architecture — fix architecture (and layer), vary sparsity; (B) Sparsity — compare architectures at matched sparsity ranks; (C) Model — fix the base model, compare all SAEs within it. For axis $i$, partition $\Theta$ into groups $\mathcal{G}_i$ that are matched on all axes except $i$. Within each group $G \in \mathcal{G}_i$, compute Kendall's coefficient in $\{(\mu(\theta), g(\theta)) : \theta \in G\}$, and average between groups to obtain the statistic at the axis level:

$$\psi_i = \frac{1}{|\mathcal{G}_i|} \sum_{G \in \mathcal{G}_i} \tau(\{(\mu(\theta), g(\theta)) : \theta \in G\}). \tag{4}$$

Aggregate the axis-level outcomes by

$$\Psi = \frac{1}{n} \sum_{i=1}^{n} \psi_i, \tag{5}$$

where $n$ is the number of axes. Each $\psi_i$ captures rank consistency conditioned on axis $i$ (varying only that axis while matching the others), and $\Psi$ aggregates these into a single axis-controlled measure. This construction mitigates cross-axis trends—e.g., architecture, sparsity, or model-driven shifts that can obscure local relationships between interpretability and utility.

We report the per-axis statistics $\psi_i$ together with the aggregate $\Psi$ for the same model settings as in section 3.2, providing both axis-specific and aggregated assessments.

## 3.4 PAIRWISE ANALYSIS RESULTS

In this section, we assess whether higher SAE interpretability predicts stronger steering by computing Kendall's $\tau_b$ between the *Interpretability Score* $\mu(\theta)$ and the *Steering Score* $g_{\text{base}}(\theta)$ (before any feature selection) over a pooled set of SAEs attached to a fixed LLM.

To control confounders and localize effects, we apply the axis-conditioned procedure defined in section 3.3. For each axis, we form matched groups, compute within-group $\tau_b$, average to obtain a per-axis summary, and aggregate these summaries into an overall axis-controlled coefficient.

Table 1: **Pairwise Analysis Between *Interpretability Score* $\mu(\theta)$ and *Steering* Score $g_{\text{base}}(\theta)$.** Here $g_{\text{base}}(\theta)$ is steering score before any feature selection. We report Kendall's $\tau_b$ overall and by axis-controlled measures $\Psi_A$ (Architecture), $\Psi_B$ (Matched Sparsity), and $\Psi_C$ (Model). $n$ = number of SAEs; Pairs = number of pairwise comparisons; $p$ = permutation $p$-value for testing $H_0 : \tau_b = 0$ within each group (by shuffling steering scores); 95% CI = 95% bootstrap percentile confidence interval for $\tau_b$.

| Axis | SAEs | $n$ | Pairs | $\tau_b$ | $p$ | 95% CI |
|------|------|-----|-------|----------|-----|--------|
| Overall | All SAEs | 90 | 4005 | **0.2979** | 0.0002 | [0.1591, 0.4191] |
| | $\Psi_A = $ **0.2575** (SE $\approx$ 0.1163, 95% boot CI [0.0222, 0.3961]) | | | | | |
| | BatchTopK | 18 | 153 | 0.3203 | 0.0712 | [0.0000, 0.6429] |
| | Gated | 18 | 153 | -0.2026 | 0.2577 | [−0.5493, 0.1918] |
| $\Psi_A$: Architecture | JumpReLU | 18 | 153 | 0.4248 | 0.0160 | [0.0563, 0.7183] |
| | ReLU | 18 | 153 | 0.3595 | 0.0392 | [0.0274, 0.6528] |
| | TopK | 18 | 153 | 0.3856 | 0.0272 | [0.0282, 0.6966] |
| | $\Psi_B = $ **0.1651** (SE $\approx$ 0.1112, 95% boot CI [−0.0286, 0.3587]) | | | | | |
| | $L_0 \approx 50$ | 15 | 105 | 0.5429 | 0.0034 | [0.1578, 0.8737] |
| | $L_0 \approx 80$ | 15 | 105 | 0.3524 | 0.0740 | [0.0000, 0.6304] |
| $\Psi_B$: Sparsity | $L_0 \approx 160$ | 15 | 105 | 0.1810 | 0.3821 | [−0.2501, 0.5464] |
| | $L_0 \approx 320$ | 15 | 105 | 0.1810 | 0.3673 | [−0.1579, 0.5208] |
| | $L_0 \approx 520$ | 15 | 105 | -0.2190 | 0.2837 | [−0.5914, 0.1648] |
| | $L_0 \approx 820$ | 15 | 105 | -0.0476 | 0.8484 | [−0.4409, 0.3750] |
| | $\Psi_C = $ **0.3272** (SE $\approx$ 0.0698, 95% boot CI [0.2184, 0.4575]) | | | | | |
| | Gemma-2-2B | 30 | 435 | 0.2184 | 0.0980 | [−0.0644, 0.4764] |
| $\Psi_C$: Model | Qwen-2.5-3B | 30 | 435 | 0.4575 | 0.0008 | [0.2086, 0.6580] |
| | Gemma-2-9B | 30 | 435 | 0.3057 | 0.0166 | [0.0521, 0.5277] |
| | $\Psi = \big(\Psi_A + \Psi_B + \Psi_C\big)/3 = $ **0.2499** | | | | | |

We complement each rank-correlation estimate with both hypothesis testing and uncertainty quantification (Table 1). Specifically, for each matched group, we report Kendall's $\tau_b$ along with a permutation test $p$-value for $H_0 : \tau_b = 0$, obtained by shuffling steering scores within the group while keeping interpretability scores fixed. We also report 95% bootstrap percentile confidence intervals (CI) for $\tau_b$ to capture sampling variability. In addition, for each axis-conditioned summary $\Psi_A, \Psi_B, \Psi_C$, we provide its standard error and a 95% bootstrap CI, and finally aggregate them into an overall axis-controlled coefficient $\Psi$.

Table 1 shows that **across SAEs, higher interpretability tends to be modestly associated with better steering on average, pointing to a consistent but limited impact.** The pooled Kendall's $\tau_b \approx 0.30$ is positive, and the axis-controlled aggregate remains positive ($\Psi \approx 0.25$), indicating that more interpretable features generally translate into better steering utility across designs and models.

**The strength of the link between interpretability and utility depends on SAE architecture, sparsity, and the base model.** By architecture, the association is positive on average ($\Psi_A \approx 0.26$), with ReLU-like variants reinforcing the trend and Gated weakening it. By sparsity, alignment is strongest when the SAE is more sparse and weakens—sometimes reversing—as the number of active features increases. By model, the underlying LM shapes the effect, with the signal clearest in Qwen-2.5-3B and weaker in Gemma-2-2B, while the model-wise summary remains positive ($\Psi_C \approx 0.33$).

> **Key Observation 1:** Interpretability shows a relatively weak positive correlation with steering performance, highlighting a notable gap between interpretability and utility across SAEs.

## 4 FROM INTERPRETABILITY TO UTILITY: WHICH SAE FEATURES ACTUALLY STEER?

In Sec. 3.4, We find that SAE interpretability is a relatively weak prior for steering utility. Prior work (Arad et al., 2025) shows many features lack steerability and we speculate that this factor may render the previous conclusion inaccurate. Therefore, we introduce a metric to identify steering-effective features. Metrics derived from a model's internal token distributions can assess reasoning

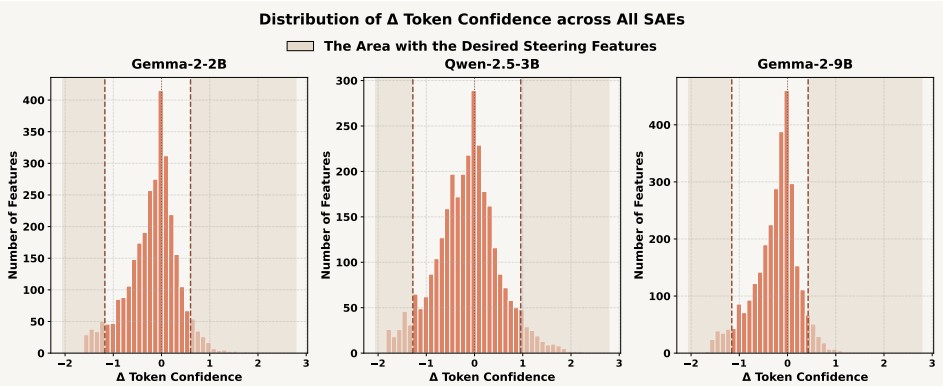

Figure 3: **Distribution of per-feature $\Delta$ *Token Confidence* across all SAEs.** Panels show histograms for Gemma-2-2B, Qwen-2.5-3B, and Gemma-2-9B; the $x$-axis is $\Delta C_k$ (negative values indicate increased confidence, positive values decreased confidence) and the $y$-axis is the number of SAE features. The shaded area marks the high-magnitude tails from which candidate steering features are selected, while the central mass near $0$ indicates features with little distributional impact.

quality (Kang et al., 2025). In particular, *token entropy* offers a unified view: high entropy highlights critical decision points (Fu et al., 2025; Wang et al., 2025a). We apply this idea to SAE steering.

## 4.1 FEATURE SELECTION VIA $\Delta$ TOKEN CONFIDENCE

We start from the model's next-token distribution. Given logits $z \in \mathbb{R}^V$ and $p = \text{softmax}(z)$ over a vocabulary of size $V$, the *token entropy* is

$$H(p) = -\sum_{j=1}^{V} p_j \log p_j, \qquad (6)$$

Entropy summarizes dispersion over the vocabulary: smaller values reflect a sharper, more concentrated prediction, while larger values indicate greater uncertainty at a given position.

To focus on the head of the distribution that matters most for sampling, we use *token confidence*. Let $\mathcal{I}_k(p) \subseteq \{1, \ldots, V\}$ denote the indices of the $k$ largest probabilities in $p$. The top-$k$ *token confidence* is the negative average log-probability over these entries:

$$C_k(p) = -\frac{1}{k} \sum_{j \in \mathcal{I}_k(p)} \log p_j. \qquad (7)$$

Lower $C_k$ implies higher confidence, while higher $C_k$ implies a flatter top-$k$ distribution. Unlike entropy, $C_k$ directly captures the sharpness of the outcomes that drive next-token behavior.

We turn confidence into a feature-level selector via a single-feature SAE intervention. Consider an SAE feature $f$ at layer $\ell$. We amplify only the coefficient of $f$ by a factor $\alpha > 0$ in the SAE reconstruction, leaving the base model and all other features unchanged. Denote the baseline next-token distribution by $p^{\text{base}}$ and the intervened distribution by $p_{f,\ell,\alpha}^{\text{int}}$. $\Delta$ *Token Confidence* is

$$\Delta C_k(f; \ell, \alpha) = C_k\big(p_{f,\ell,\alpha}^{\text{int}}\big) - C_k\big(p^{\text{base}}\big). \qquad (8)$$

Negative values $\Delta C_k < 0$ mean that amplifying $f$ sharpens the top-$k$ distribution, while positive values indicate greater dispersion. We compute this using one baseline and one intervened forward pass via an SAE hook. Implementation details and hyperparameters are provided in Appendix D.

We select features with extreme signed changes in *token confidence* under single-feature SAE interventions, and pick the better direction (see Figure 3). For each feature, we compute $\Delta C_k$, rank by $|\Delta C_k|$, form tiers, evaluate subsets for steering, and keep the best per SAE.

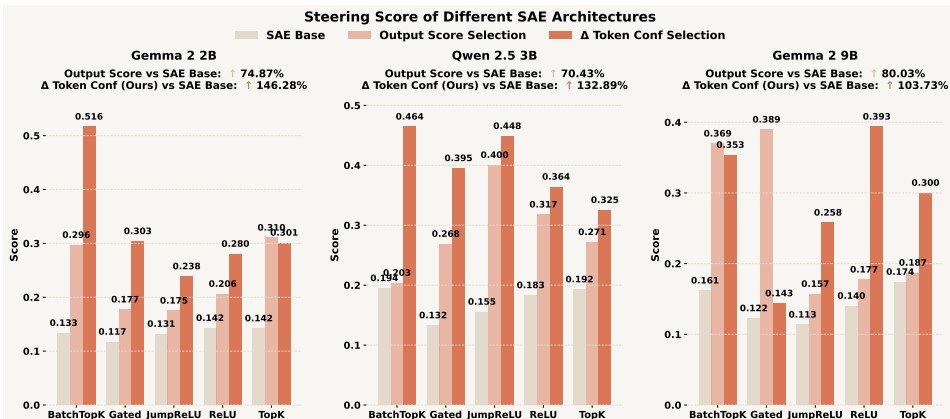

Figure 4: **Comparison of different SAE steering methods with five SAE architecture across three LLMs.** Panels correspond to Gemma-2-2B, Qwen-2.5-3B, and Gemma-2-9B. The horizontal axis groups SAE architectures (BatchTopK, Gated, JumpReLU, ReLU, TopK), and the vertical axis reports the *steering score*. Bars show three conditions: SAE Base (no feature selection), Output Score Selection, and $\Delta$ *Token Confidence* Selection (ours). Panel annotations summarize the average lift of each selection method relative to the SAE-based steering.

## 4.2 Steering Performance Results After Feature Selection

Arad et al. (2025) has shown that SAE steering works well if features are chosen by their causal impact on model outputs, introducing the *output score* as a metric to identify output-aligned features. For a given SAE feature, a logit-lens procedure is used to select a representative token set $M$. Let $P_{\text{base}}(M)$ and $P_{\text{int}}(M)$ denote the aggregate support for $M$ in the base and intervened (feature-amplified) runs, respectively, where

$$P(M) = \left(1 - \frac{\min_{i \in M} \text{rank}(i)}{|V|}\right) \max_{i \in M} p(i),$$

with $|V|$ the vocabulary size, $\text{rank}(i)$ the rank of token $i$, and $p(i)$ its probability. The single-feature steering score is then

$$S_{\text{out}} = P_{\text{int}}(M) - P_{\text{base}}(M),$$

which is large when amplifying the feature increases the rank and probability of representative token set. Following this insight, we evaluate our $\Delta$ *token confidence* selection on three base LLMs (Gemma-2-2B, Qwen-2.5-3B, Gemma-2-9B) using the CONCEPT100 (see details in 3.1). We set $k = 1$ in our $\Delta$ *token confidence* selection. In ablations over $k \in \{1, 3, 5, 10\}$ on Gemma and Qwen, all choices of $k$ yield large gains over the SAE baseline, and $k = 1$ is in fact the best-performing setting on both models, so we adopt it as our default (see Appendix L). The experiments on steering performance improvement of each SAE can be referred to Appendix E.

Table 2 shows that our selection yields consistent gains across all models, outperforming the vanilla SAE baseline by large margins, and also improving over an output-score–based selector. These gains indicate that ranking and filtering by the magnitude of distributional change captured by $\Delta C_k$ reliably isolates features with the strongest steering utility.

Furthermore, we conducted a comparative analysis of SAEs of different architectures on three models. For fair comparison, the two feature selection methods use the same subset size. Figure 4 compares *steering scores* across SAE architectures and

Table 2: *Steering score* **after feature selection compared with SAE-based steering.** Columns report scores (higher is better) for Gemma-2-2B, Qwen-2.5-3B, and Gemma-2-9B. Rows: 'SAE-based' uses all SAE features without selection (Wu et al., 2025); '+Output' selects features using $S_{\text{out}}$ (Arad et al., 2025); "+$\Delta C_k$ (Ours)" selects by the $\Delta$ *Token Confidence*. Boldface indicates the best method per model.

| Method | Gemma-2-2B | Qwen-2.5-3B | Gemma-2-9B |
|---|---|---|---|
| SAE-based | 0.133 | 0.171 | 0.142 |
| +Output | 0.233 | 0.292 | 0.255 |
| +$\Delta C_k$(Ours) | **0.328** | **0.399** | **0.289** |

selection methods. In all three models, selecting features by $\Delta$ *Token Confidence* consistently outperforms both the no-selection SAE baseline and the output-score selector across architectures.

On average, our method improves steering performance by **52.52%** over the strongest competing baseline. The BatchTopK architecture is the one that has the most stable and significant improvement in steering capabilities on models of different sizes among the five SAE architectures.

> **Key Observation 2:** $\Delta$ *Token Confidence* reliably selects high-utility SAE features across models. Among SAE architectures, BatchTopK achieves the most stable and sizable *steering score*.

### 4.3 PAIRWISE ANALYSIS AFTER HIGH-CONFIDENCE FEATURE SELECTION

Building on the high $\Delta$ *Token Confidence* feature selection introduced above, we now ask whether SAE interpretability can serve as a prior for *post-selection steering performance*. For each SAE $\theta$, we write $g_{\text{base}}(\theta)$ for its *Steering Score* before feature selection and $g_{\text{high}}(\theta)$ for the best score achieved after selecting features with high $\Delta$ *Token Confidence*. Both quantities are measured in the same steering metric; $g_{\text{high}}(\theta)$ is simply the steering score recomputed on a restricted set of high $\Delta$ *Token Confidence* features for the same SAE.

We quantify the relationship between interpretability and post-selection steering by computing Kendall's $\tau_b$ between the *Interpretability Score* $\mu(\theta)$ of SAEs and the *Steering* Score $g_{\text{high}}(\theta)$. As in Section 3.4, we report both pooled coefficients and axis-conditioned summaries that control for design and model factors (architecture, sparsity and model). In addition, Section 3.4 and Table 1 report the corresponding coefficients for $\mu(\theta)$ vs. $g_{\text{base}}(\theta)$, allowing a direct comparison between "before" and "after" feature selection using the same steering metric.

Overall, Table 3 shows that interpretability is not a reliable prior for steering performance after selection. The pooled association between $\mu(\theta)$ and $g_{\text{high}}(\theta)$ is small and statistically indistinguishable from zero ($\tau_b \approx 0.08$), and the axis-controlled aggregate is likewise close to zero ($\Psi \approx 0.07$). Estimates across architectures, sparsity levels, and models cluster around zero and are mostly non-significant. These results indicate that once we focus on features that are most useful for steering (as selected by high $\Delta$ *Token Confidence*), higher interpretability can't predict better steering scores.

> **Key Observation 3:** Surprisingly, the interpretability–utility gap widens after feature selection: higher interpretability scores can't provide a prior for which SAEs achieve better steering.

## 5 RELATED WORK

### 5.1 REPRESENTATION-BASED STEERING

Activation-based steering arose as a lightweight alternative to fine-tuning, enabling on-the-fly control of LLM behavior without retraining (Giulianelli et al., 2018; Vig et al., 2020; Geiger et al., 2021; 2025). The core idea is to inject carefully chosen directions into hidden states, typically in the residual stream, scaling interventions by a gain and selecting layers for maximal effect (Zou et al., 2025; Rimsky et al., 2024; van der Weij et al., 2024). It has been applied to safety and moderation, persona and sentiment control, and instruction adherence, promising low-latency deployment-time adjustment but facing polysemantic entanglement and brittleness that motivate standardized evaluation (Chen et al., 2025; Liu et al., 2024). However, this approach injects polysemantic activations at intervention time, yielding coarse-grained effects for output control (Bricken et al., 2023). Our work is related to activation-level interventions, but differs by grounding directions in sparse, interpretable SAE features and applying utility-oriented feature selection to mitigate these failure modes.

### 5.2 SAE-BASED STEERING

Sparse Autoencoders (SAEs) decompose activations into sparse, human-readable features to mitigate polysemanticity and expose concept-level structure (Bricken et al., 2023; Templeton et al., 2024; Gao et al., 2024). For steering, practitioners use decoder atoms as directions and add scaled injections at chosen layers, with architecture and sparsity choices trading reconstruction for feature

Table 3: **Pairwise Analysis Between *Interpretability Score* $\mu(\theta)$ and *Steering Score* $g_{high}(\theta)$.** Here $g_{high}(\theta)$ is steering score after selecting features with high $\Delta$ Token Confidence. We report Kendall's $\tau_b$ overall and under axis-controlled summaries $\Psi_A$ (Architecture), $\Psi_B$ (Matched Sparsity), and $\Psi_C$ (Model). $n$ = number of SAEs; Pairs = number of pairwise comparisons; $p$ = permutation $p$-value for testing $H_0 : \tau_b = 0$ within each group (by shuffling steering scores); 95% CI = 95% bootstrap percentile confidence interval for $\tau_b$.

| Axis | SAEs | $n$ | Pairs | $\tau_b$ | $p$ | 95% CI |
|---|---|---|---|---|---|---|
| Overall | All SAEs | 90 | 4005 | **0.0823** | 0.2645 | $[-0.0560, 0.2243]$ |
| | $\Psi_A = \mathbf{0.0618}$ (SE $\approx$ 0.1136, 95% boot CI $[-0.1149, 0.2810]$) | | | | | |
| | BatchTopK | 18 | 153 | -0.1895 | 0.2933 | $[-0.5694, 0.1918]$ |
| | Gated | 18 | 153 | 0.0338 | 0.8764 | $[-0.3656, 0.3853]$ |
| $\Psi_A$: Architecture | JumpReLU | 18 | 153 | 0.0950 | 0.6209 | $[-0.3373, 0.4697]$ |
| | ReLU | 18 | 153 | -0.1007 | 0.5803 | $[-0.4833, 0.2857]$ |
| | TopK | 18 | 153 | 0.4702 | 0.0078 | $[0.0794, 0.8043]$ |
| | $\Psi_B = \mathbf{0.1002}$ (SE $\approx$ 0.0700, 95% boot CI $[-0.0288, 0.2188]$) | | | | | |
| | $L_0 \approx 50$ | 15 | 105 | 0.2488 | 0.2212 | $[-0.1828, 0.6300]$ |
| | $L_0 \approx 80$ | 15 | 105 | 0.2899 | 0.1602 | $[-0.0673, 0.5876]$ |
| $\Psi_B$: Sparsity | $L_0 \approx 160$ | 15 | 105 | 0.1183 | 0.5747 | $[-0.2357, 0.4151]$ |
| | $L_0 \approx 320$ | 15 | 105 | 0.1171 | 0.5803 | $[-0.3531, 0.5209]$ |
| | $L_0 \approx 520$ | 15 | 105 | -0.1827 | 0.3747 | $[-0.6490, 0.3524]$ |
| | $L_0 \approx 820$ | 15 | 105 | 0.0097 | 1.0000 | $[-0.3724, 0.3572]$ |
| | $\Psi_C = \mathbf{0.0538}$ (SE $\approx$ 0.0700, 95% boot CI $[-0.0540, 0.1850]$) | | | | | |
| | Gemma-2-2B | 30 | 435 | -0.0540 | 0.6963 | $[-0.3127, 0.2125]$ |
| $\Psi_C$: Model | Qwen-2.5-3B | 30 | 435 | 0.1850 | 0.1668 | $[-0.0872, 0.4413]$ |
| | Gemma-2-9B | 30 | 435 | 0.0303 | 0.8284 | $[-0.2552, 0.3233]$ |
| | $\Psi = (\Psi_A + \Psi_B + \Psi_C)/3 = \mathbf{0.0719}$ | | | | | |

granularity (Zhao et al., 2025; Wang et al., 2025d; Ferrando et al., 2025). SAE-based steering enables targeted safety control, style modulation, and instruction emphasis, yet the utility of individual features varies widely (Chalnev et al., 2024; Mayne et al., 2024). While the connection between SAE interpretability and steering utility remains unclear, and our goal is to build a principled bridge between them. To this end, we conduct a large-scale experiments across multiple model sizes and SAE architectures, demonstrating the critical nature of the interpretability-utility gap.

# 6 CONCLUSION AND DISCUSSION

We study a focused but practically important question: *within the SAE paradigm*, does higher *SAE feature interpretability* predict better *steering utility*? Across 90 SAEs spanning three base LLMs, five architectures, and six sparsity levels, interpretability shows only a weak positive association with steering utility ($\tau_b \approx 0.298$), revealing a clear interpretability–utility gap. Feature selection with $\Delta$ *Token Confidence* substantially improves steering, yet among the highest-utility features the correlation collapses toward zero and can even turn negative, suggesting that interpretability and steerability diverge most in the regime that matters for control.

A key practical implication is that *SAE interpretability scores are hard to use as a reliable indicator for downstream performance in SAE-based applications*. Optimizing or selecting SAEs purely by interpretability can mis-rank models when the goal is controllable generation, and utility-aware selection or training objectives are often necessary. Looking forward, narrowing this gap likely requires either (i) post-training protocols that identify causally effective features, or (ii) utility-oriented SAE training paradigms that directly optimize controllability under sparsity.

Finally, *SAE feature explanations remain highly valuable*: we expect human-readable SAE descriptions to support broader workflows such as data filtering, targeted data synthesis, and concept-level steering. Our results therefore do not diminish the promise of SAE explanations; they clarify that current interpretability scores alone are insufficient for selecting SAEs for downstream use.

ACKNOWLEDGMENTS

We would like to thank the anonymous reviewers and area chairs for their helpful comments. We acknowledge the support from NSFC 62306252, Hong Kong ECS award 27309624, Guangdong NSF 2024A1515012444, and the International Science and Technology Cooperation Center, Ministry of Science and Technology of China (under grant 2024YFE0203000).

REPRODUCIBILITY STATEMENT

We aim to facilitate full reproduction of our results. All model code, training and evaluation scripts, and experiment logs are released at the repository as part of the supplementary materials: https://github.com/Xu0615/SAE4Steer. Training architectures, hyperparameters, sparsity schedules, and optimization details are specified in the main text and Appendix A.2 (see also the per-family settings in Appendix A). The datasets used are openly licensed: all SAEs are trained on The Common Pile v0.1 (Kandpal et al., 2025) as described in Appendix F; our evaluation concepts (CONCEPT100) and their automatic generation pipeline are documented in Appendix B and Appendix F. The complete procedures for automated interpretability scoring (SAEBENCH) and steering utility (AXBENCH), including sampling, judging protocols, and scoring functions, are detailed in Appendix B and Appendix C, with the $\Delta$ *Token Confidence* selector defined in Appendix D and the post-selection results summarized in Appendix E. Hardware, runtime, and memory footprints for both SAEBENCH and AXBENCH are reported in Appendices B.3 and C.2. Together, these materials, along with seed-controlled configuration files and exact command-line invocations provided in the repository, are intended to enable independent researchers to replicate and extend our findings.

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

## LLM USAGE

In preparing this paper, large language models (LLMs) were used as an assistive tool for minor language polishing and stylistic improvements. All technical contributions, results, and conclusions are solely the work of the authors.

## A  SAE ARCHITECTURES AND TRAINING DETAILS

We train **90 SAEs** (30 per base model) across five architectures and six target sparsity levels. Unless stated otherwise, the dictionary width is **16K** codes ($F=16{,}384$), SAEs are attached to the residual stream at the layer described in the main text, and decoder columns are $\ell_2$–normalized. All models are trained on The Common Pile v0.1 (Kandpal et al., 2025).

### A.1  ARCHITECTURES AND PARAMETERIZATION

We list the five SAE families with their named parameters (as implemented) and the corresponding shapes. The last column records architecture-specific thresholding/gating fields when present. Shapes assume residual dimension $d=2304$ and dictionary width $F=16{,}384$.

| Architectures | $W_{\mathbf{enc}}$ | $b_{\mathbf{enc}}$ | $W_{\mathbf{dec}}$ | $b_{\mathbf{dec}}$ | Threshold / Extras |
|---|---|---|---|---|---|
| ReLU | encoder.weight: shape (16,384, 2,304) | encoder.bias: shape (16,384) | decoder.weight: shape (2,304, 16,384) | bias: shape (2,304) | — |
| Gated | encoder.weight: shape (16,384, 2,304) | gate_bias: shape (16,384) | decoder.weight: shape (2,304, 16,384) | decoder_bias: shape (2,304) | r_mag: shape (16,384); mag_bias: shape (16,384) |
| TopK | encoder.weight: shape (16,384, 2,304) | encoder.bias: shape (16,384) | decoder.weight: shape (2,304, 16,384) | b_dec: shape (2,304) | k |
| BatchTopK | encoder.weight: shape (16,384, 2,304) | encoder.bias: shape (16,384) | decoder.weight: shape (2,304, 16,384) | b_dec: shape (2,304) | k |
| JumpReLU | W_enc: shape (2,304, 16,384) | b_enc: shape (16,384) | W_dec: shape (16,384, 2,304) | b_dec: shape (2,304) | threshold: shape (16,384) |

### A.2 Training, Sparsity, and Compute Setup

**Optimization and schedule.** Adam with learning rate $3 \times 10^{-4}$; LR warmup 1000 steps; sparsity warmup 5000 steps; LR decay starting at $80\%$ of total steps. Precision: `bfloat16`. LM batch size $= 4$, context length $= 2048$, SAE batch size $= 2048$. Each run trains on $\sim 5 \times 10^8$ tokens.

**Sparsity controls.** We sweep six target activity levels

$$L_0 \approx \{50, 80, 160, 320, 520, 820\}.$$

For TopK/BatchTopK we set $k$ equal to the chosen $L_0$ (aux-$k$ coefficient $1/32$; moving-threshold momentum $0.999$; threshold tracking begins at step 1000). JumpReLU uses the same set via `target_l0`. For $L_1$–penalized families, we search the following penalty grids:

| Family | $L_1$ **penalty values (used to span sparsity levels)** |
|---|---|
| Standard / Standard-New | 0.012, 0.015, 0.020, 0.030, 0.040, 0.060 |
| Gated SAE | 0.012, 0.018, 0.024, 0.040, 0.060, 0.080 |

**Training details.** All training uses two NVIDIA RTX A800 GPUs. The table below reports the aggregated artifacts and training time (hours) for *30 SAEs per model* (total 90), together with the runtime configuration. Times and sizes are approximate.

| Model | #SAEs | Disk (GB) | Traing Time (H) | LM Batch | Context | SAE Batch | Peak Mem (GB) |
|---|---|---|---|---|---|---|---|
| Gemma-2-2B | 30 | 8.7 | 17 | 4 | 2048 | 2048 | 20 |
| Gemma-2-9B | 30 | 13.2 | 60 | 4 | 2048 | 2048 | 70 |
| Qwen-2.5-3B | 30 | 7.7 | 37 | 4 | 2048 | 2048 | 30 |

## B SAEBench Details, Results and Our Costs

### B.1 Automated Interpretability Score Process

SAEBench (Karvonen et al., 2025) follow an LLM-as-judge pipeline to assign an *automated interpretability score* to each SAE latent. First, we collect layer activations by running the base LM with caching and encoding the residual stream through the SAE to obtain $h \in \mathbb{R}^{N \times L \times F}$. We define a token window of length 21 (buffer $= 10$) around any center $(i, t)$ and, unless stated otherwise, mask BOS/PAD/EOS positions. For a latent $\ell$, we sample three window types: (i) **Top** ($n = 12$ non-overlapping peaks of $h[:, :, \ell]$), (ii) **Importance-Weighted** ($n = 7$, sampled proportional to activation after removing values at least as large as the smallest Top peak), and (iii) **Random** ($n = 10$, uniform over valid centers). Let $v_{\max}$ be the maximum activation seen in any Top window position and set a global threshold $\tau_{\text{act}} = 0.01\, v_{\max}$.

We split the sampled windows into a **generation set** (10 Top + 5 IW) and a **scoring set** (2 Top + 2 IW + 10 Random, shuffled). In generation, tokens with activation $> \tau_{\text{act}}$ are bracketed to highlight evidence; the judge LLM receives these 15 windows and returns a short English description of when the latent fires. In scoring, the judge sees the description and the 14 held-out windows *without* highlights and outputs a comma-separated list of indices it predicts as activations (or `None`).

Ground truth for a window $\mathcal{W}$ is $\mathbb{1}[\max_{u \in \mathcal{W}} h[u, \ell] > \tau_{\text{act}}]$. The per-latent score is the **accuracy** over the $M = 14$ scoring windows, i.e.,

$$\text{Score}(\ell) = \frac{1}{M} \sum_{m=1}^{M} \mathbf{1}[\widehat{y}_m = y_m],$$

where $\widehat{y}_m \in \{0,1\}$ is the judge prediction and $y_m$ is the label defined above. For each SAE $\theta$, we evaluate $1{,}000$ latents and report the mean over a random CONCEPT100 subset:

$$\mu(\theta) = \frac{1}{100} \sum_{\ell \in \text{CONCEPT100}} \text{Score}(\ell).$$

## B.2 PERFORMANCE OF SAEs ON THREE MODELS ON SAEBENCH

Across the three backbones, the six SAEBENCH metrics (for information about these indicators, see SAEBENCH (Karvonen et al., 2025))jointly reveal how sparsity mechanisms balance interpretability, faithfulness, and causal structure. Automated Interpretability is strongest when encoders enforce compact latent usage (e.g., TopK/BatchTopK and ReLU at lower $L_0$), and it gradually softens as capacity expands. The Absorption metric (considered via its complement in the plots) indicates that designs concentrating signal into a small set of latents are less prone to feature stealing, whereas higher effective capacity encourages redundancy and competition across latents. Meanwhile, Core/Loss-Recovered remains uniformly high, showing that even sparse codes closely preserve original model behavior; increasing $L_0$ pushes faithfulness toward a ceiling without overturning the core trade-offs visible in the other metrics.

**Gemma-2-2B.** As shown in Fig. 5, Gemma-2-2B exhibits a balanced profile: interpretability stays robust for TopK/BatchTopK and ReLU at modest sparsity; absorption is contained when the code remains compact; and Core is near-saturated across the range. Improvements in SCR@20 are steady but measured, suggesting targeted debiasing with small $k$. Sparse Probing indicates that relatively few latents already carry much of the predictive signal, while RAVEL strengthens with moderate capacity, reflecting cleaner separation of attributes without undermining compactness.

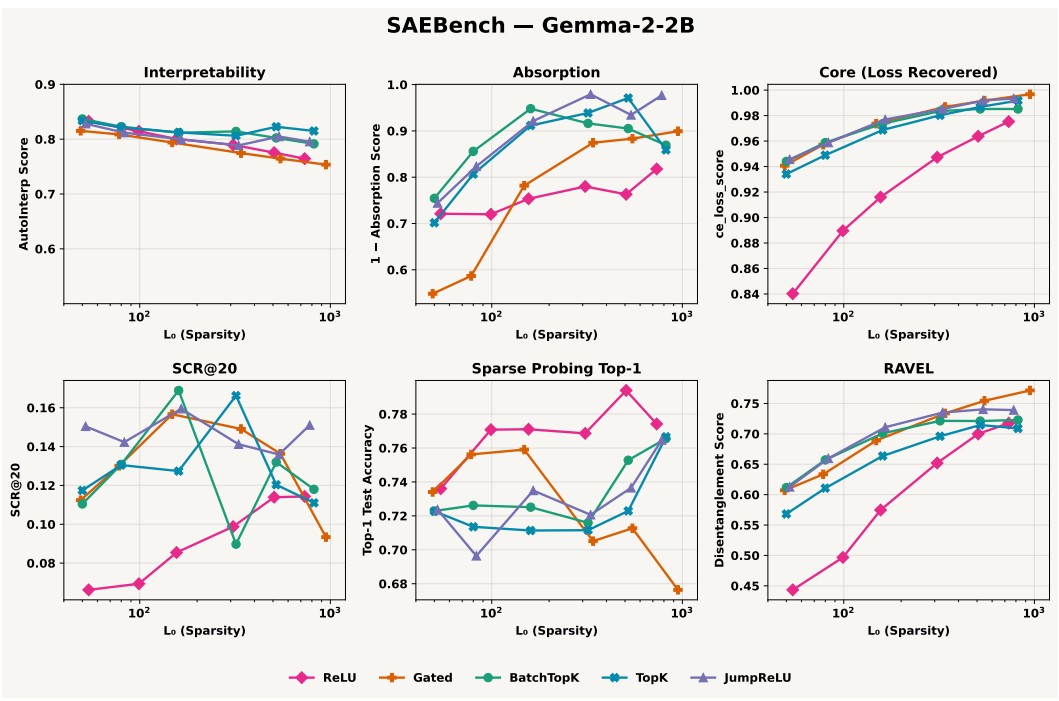

Figure 5: SAEbench results for **Gemma-2-2B**: interpretability remains strong at lower $L_0$, absorption stays low for compact codes, Core is near ceiling, and structure (SCR/RAVEL) improves with moderate capacity.

**Qwen-2.5-3B.** For Qwen-2.5-3B (Fig. 6), interpretability at low–to–moderate $L_0$ is competitive—especially for TopK and JumpReLU—yet the model is more sensitive to absorption as capacity grows, implying greater latent competition and signal spread. Core remains excellent, so

reconstructions are faithful; however, SCR gains can flatten at high $L_0$ where residual spurious cues reappear. Sparse Probing is solid but a touch behind the strongest Gemma configurations, consistent with its flatter RAVEL patterns: causal structure is present but less crisply disentangled when attributes begin to diffuse across latents.

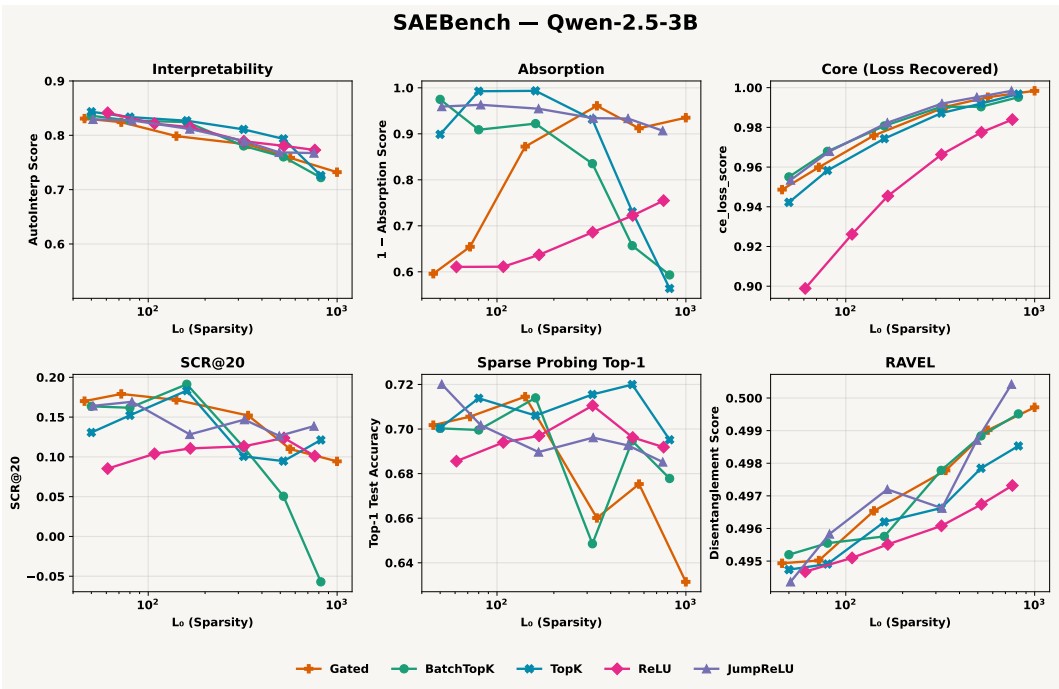

Figure 6: SAEbench results for **Qwen-2.5-3B**: strong interpretability at lower $L_0$, rising absorption with capacity, consistently high Core, and more fragile SCR/RAVEL at the highest capacities.

**Gemma-2-9B.** Gemma-2-9B (Fig. 7) pushes the upper envelope on structure: interpretability remains solid for compact encoders; absorption is low at moderate $L_0$ that avoids unnecessary latent proliferation; and Core is near its ceiling. SCR@20 is the most decisive among the three, pointing to cleaner isolation of spurious factors with small, targeted ablations. Sparse Probing is strong and, together with higher RAVEL, indicates that only a handful of latents capture both predictive signal and causally specific attributes with minimal collateral interference.

### B.3 SAEBENCH RUNTIME COST

The computational requirements for running SAEBench evaluations were measured on two NVIDIA RTX A800 GPUs using **16K**-width SAEs trained on the Gemma-2-2B (Team et al., 2024), Qwen-2.5-3B (Yang et al., 2024) and Gemma-2-9B. Table 4 summarizes the *per-SAE* runtime for each evaluation type. Several evaluations include a one-time setup phase (e.g., precomputing activations or training probes) that can be reused across multiple SAEs; after this setup, each evaluation has its own runtime per SAE. We therefore report amortized per-SAE minutes.

Table 4: **Approximate SAEBench runtime per SAE (minutes).** Values are per-SAE and represent amortized minutes after any one-time setup; each minute figure is an approximation and may vary with hardware and I/O.

| Model | Core | Interpretability | Absorption | Sparse Probing | Ravel | SCR |
|---|---|---|---|---|---|---|
| Gemma-2-2B | 4 | 8 | 12 | 2 | 18 | 10 |
| Qwen-2.5-3B | 7 | 9 | 15 | 8 | 17 | 16 |
| Gemma-2-9B | 11 | 12 | 17 | 30 | 40 | 28 |

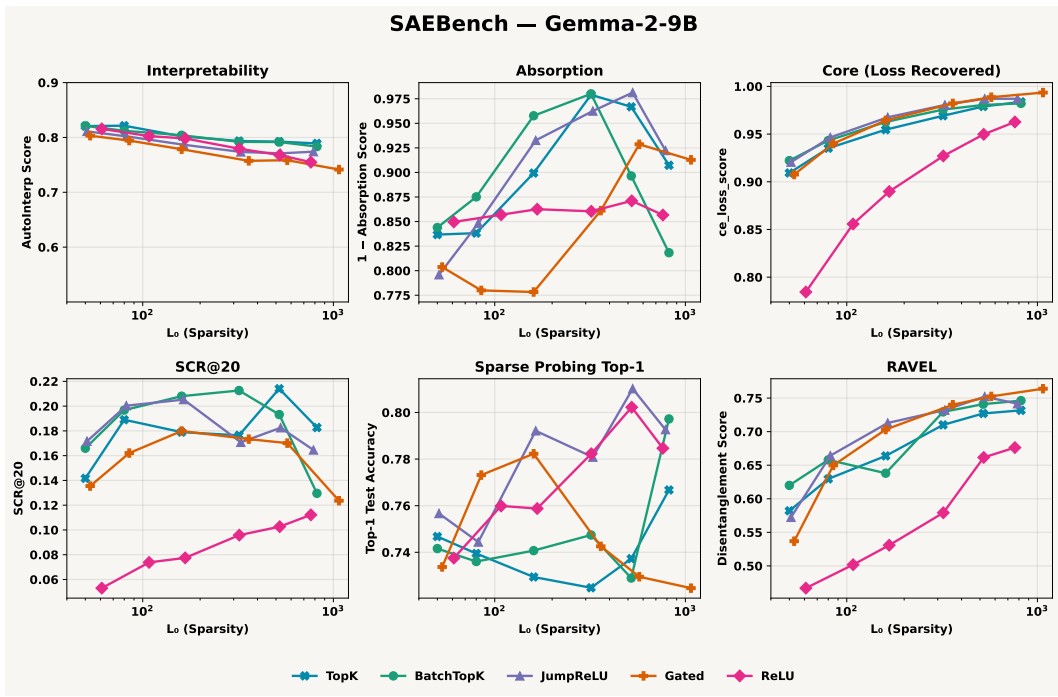

Figure 7: SAEbench results for **Gemma-2-9B**: robust interpretability with compact codes, low absorption at moderate $L_0$, near-ceiling Core, and the clearest gains in SCR/RAVEL among the three backbones.

## C  AXBENCH DETAILS AND OUR COSTS

### C.1  STEERING SCORE PROCESS

For each SAE feature (concept) we evaluate steering utility with an AXBENCH-style judge (Wu et al., 2025). Given the concept description from SAEBench and the intervention rule in Eq. equation 1, we sample 10 instructions (Alpaca-Eval style) and split them $5/5$ into a development set for factor selection and a held-out set for evaluation. We generate continuations (up to 128 tokens) for a grid of steering factors $\alpha \in \mathcal{A}$ by injecting the feature direction at the target layer. For each generated continuation, an LLM judge assigns three subscores on the discrete scale $\{0, 1, 2\}$ (floats allowed but clamped to $[0, 2]$): *Concept* ($C$; does the output express the concept?), *Instruction* ($I$; is it on-topic for the instruction?), and *Fluency* ($F$; is it well-formed?). We aggregate with the harmonic mean

$$\text{HM}(C, I, F) = \begin{cases} 0, & \text{if } C{\leq}0 \text{ or } I{\leq}0 \text{ or } F{\leq}0, \\ \dfrac{3}{\frac{1}{C} + \frac{1}{I} + \frac{1}{F}}, & \text{otherwise,} \end{cases} \in [0, 2].$$

On the development split we pick $\alpha^\star = \arg\max_{\alpha \in \mathcal{A}} \frac{1}{5} \sum \text{HM}$, then evaluate the held-out split with $\alpha^\star$ to obtain the *per-concept Steering Score* (mean HM over the 5 held-out prompts). SAE-level utility $g(\theta)$ is the average of per-concept scores over the evaluation set for that SAE.

**CONCEPT_TEMPLATE**
*[System]*
Please act as an impartial judge and evaluate whether the specified concept is presented in the sentence fragment provided below. Focus solely on whether the concept is clearly incorporated, without regard for grammar or logical coherence.
Begin your evaluation by providing a short explanation of whether the concept is clearly incorporated. Point out references to the concept in the fragment, noting any minimal or awkward incorporations. After providing your explanation, rate the concept's relevance on a scale from 0 to 2, where 0 indicates the

concept is not present at all, 1 indicates the concept is somewhat present but minimally or awkwardly incorporated, and 2 indicates the concept is more fully and effectively incorporated, with stronger and more natural integration. Provide your rating using this exact format: "Rating: [[score]]".

*[Concept Start]* {concept} *[Concept End]*
*[Sentence Fragment Start]* {sentence} *[Sentence Fragment End]*

**INSTRUCT_TEMPLATE**
*[System]*
Please act as an impartial judge and evaluate whether the sentence fragment provided below is related to the instruction. Focus solely on the degree of relatedness in terms of topic, regardless of grammar, coherence, or informativeness.
Begin your evaluation by providing a brief explanation of whether the sentence is related to the instruction, and point out references related to the instruction. After providing your explanation, rate the instruction relevance on a scale from 0 to 2, where 0 indicates the sentence is unrelated to the instruction, 1 indicates it is somewhat related but only minimally or indirectly relevant in terms of topic, and 2 indicates it is more clearly and directly related to the instruction. Provide your rating using this exact format: "Rating: [[score]]".

*[Instruction Start]* {instruction} *[Instruction End]*
*[Sentence Fragment Start]* {sentence} *[Sentence Fragment End]*

**FLUENCY_TEMPLATE**
*[System]*
Please act as an impartial judge and evaluate the fluency of the sentence fragment provided below. Focus solely on fluency, disregarding its completeness, relevance, coherence with any broader context, or informativeness.
Begin your evaluation by briefly describing the fluency of the sentence, noting any unnatural phrasing, awkward transitions, grammatical errors, or repetitive structures that may hinder readability. After providing your explanation, rate the sentence's fluency on a scale from 0 to 2, where 0 indicates the sentence is not fluent and highly unnatural (e.g., incomprehensible or repetitive), 1 indicates it is somewhat fluent but contains noticeable errors or awkward phrasing, and 2 indicates the sentence is fluent and almost perfect. Provide your rating using this exact format: "Rating: [[score]]".

*[Sentence Fragment Start]* {sentence} *[Sentence Fragment End]*

## C.2 AxBench Steering Evaluation Cost

All steering-score evaluations were run on **two NVIDIA RTX A800 GPUs**. The LLM-as-judge backend was `gpt-4o-mini`. Evaluating one SAE on CONCEPT100 costs approximately **\$5** in judge API fees; with **90** SAEs total ($\approx$ 30 per model), the per-model API cost is about **\$150**. Table 5 lists approximate per-SAE runtime and peak VRAM for each model.

Table 5: **AxBench steering evaluation cost per SAE.** Runtimes are per-SAE (hours) and approximate; VRAM is peak memory (GB). Judge fees assume `gpt-4o-mini`: $\sim$ \$5 per SAE on CONCEPT100; -Model Cost assumes $\sim$ 30 SAEs/model ($\approx$ \$150).

| Model | Runtime / SAE (h) | Peak VRAM (GB) | Per-Model Cost (USD) |
|---|---|---|---|
| Gemma-2-2B | 15 | 10 | 150 |
| Qwen-2.5-3B | 16 | 12 | 150 |
| Gemma-2-9B | 23 | 36 | 150 |

## D Implementation of $\Delta$ Token Confidence

For a fixed, neutral prefix $s$ (we use *"From my experience,"*, following the previous work(Arad et al., 2025)) we compare the next-token distribution of the base model with that of an intervened model in which a single SAE feature is amplified at layer $L$ by a factor $\alpha$. The intervention is applied via the same SAE hook point used during training (on the residual stream of block $L$). We then compute the change in a confidence surrogate built from the top-$k$ probabilities.

**Token confidence. (Fu et al., 2025)** For a distribution $p$ over the vocabulary, let $p_{(1)} \geq \cdots \geq p_{(k)}$ be the top-$k$ probabilities.

$$C_k(p) = -\frac{1}{k} \sum_{i=1}^{k} \log p_{(i)}.$$

**Delta token confidence.** With $p_{\text{base}}$ from a standard forward pass and $p_{\text{int}}$ from a pass with the SAE feature intervention,

$$\Delta C_k(f; \alpha, L) = C_k(p_{\text{int}}) - C_k(p_{\text{base}}).$$

Each feature $f$ is evaluated with two single-step forwards on the same prefix $s$: (i) a baseline pass; (ii) an intervened pass where we scale the code for $f$ by $\alpha$ before decoding it into the residual at layer $L$ while keeping all other codes at zero. Hooks are removed immediately after the intervened pass to prevent accumulation across evaluations. In this work, we choose $\alpha = 10$ and $k = 1$.

**Feature selection from $\Delta C_k$.** For each SAE we rank its features by $\Delta C_k$ in two directions: *UP* (largest positive $\Delta C_k$) and *DOWN* (most negative $\Delta C_k$). We form selection sets using either (i) top-$K$ by magnitude with $K \in \{1, 2, 3, 4, 5\}$ per direction, or (ii) upper/lower-tail quantiles (e.g., $q \in \{0.99, 0.95, 0.90, 0.80\}$ mirrored for the lower tail). These sets are then carried into AXBENCH (Wu et al., 2025) to measure utility lift.

## E   STEERING RESULTS OF SAE ARCHITECTURES AFTER FEATURE SELECTION

We quantify steering with the AXBENCH judge after selecting features using $\Delta$ *Token Confidence* (Appendix D). Unless otherwise noted, lifts are reported as the percentage change of a given SAE's *steering score* relative to its own baseline (no selection). Results are organized at three levels: aggregate across SAEs per base model, per-SAE rankings, and distribution by architecture.

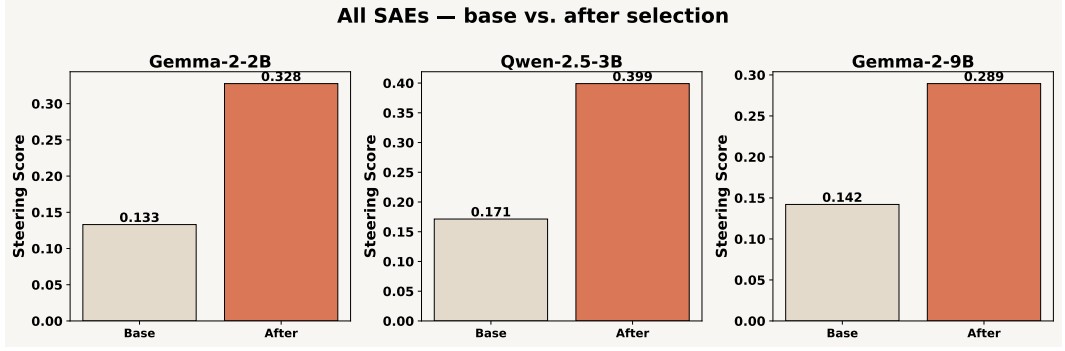

Figure 8: **Overall *steering score* before and after feature selection.** For each base model, the panel shows two bars: the average baseline *steering score* across its SAEs and the average after applying $\Delta$ Token Confidence–based selection. Bars are annotated with the corresponding values; axes share the same scale across panels.

The aggregate view in Figure 8 summarizes how selection affects the mean *steering score* across all SAEs of a base model. Using $\Delta$ *Token Confidence* for feature selection markedly improves the *steering score* across all three models in the figure. For Gemma-2-2B, the score rises from 0.133 to 0.328, which is a 146.6% relative improvement. Qwen-2.5-3B increases from 0.171 to 0.399, a 133.3% improvement, and achieves the highest post-selection score overall. Gemma-2-9B moves from 0.142 to 0.289, a 103.5% improvement.

**Conclusions:** (i) feature selection via $\Delta$ *Token Confidence* consistently boosts steering for all models; (ii) relative gains are largest for the smallest model (Gemma-2-2B) and smallest for the largest model (Gemma-2-9B), suggesting diminishing relative returns with scale; and (iii) in absolute terms, Qwen-2.5-3B reaches the strongest final *steering score* after selection.

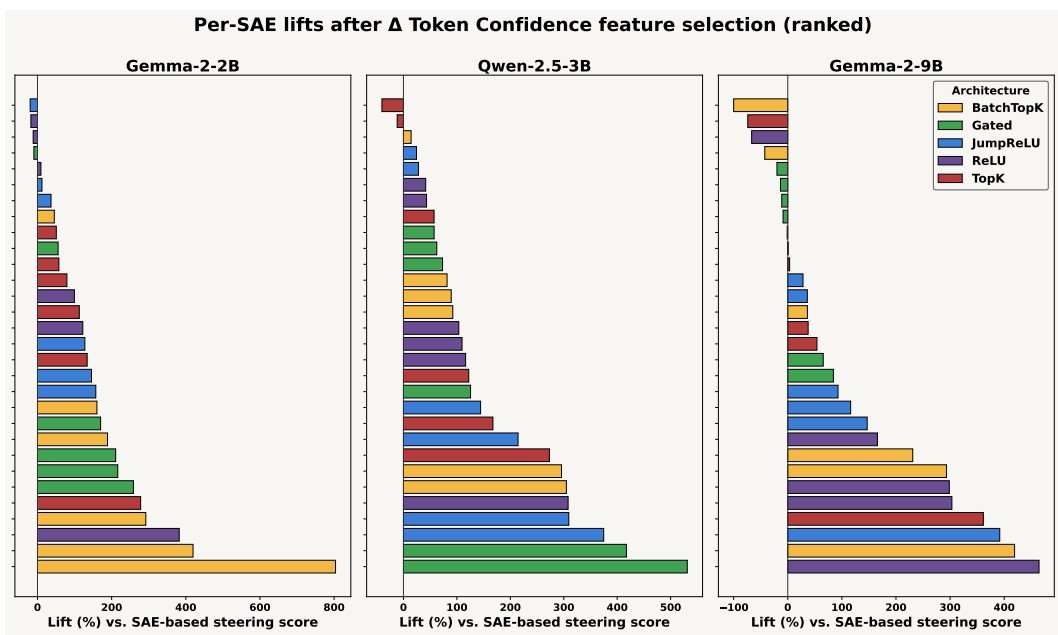

Figure 9: **Per-SAE percentage lift after Δ Token Confidence selection.** Each panel corresponds to a base model. Horizontal bars report the percent lift of the SAE-level *steering score* relative to its own baseline, sorted from largest to smallest within the panel. Bar colors indicate the SAE training architecture (legend shared across panels).

Figure 9 ranks SAEs within each model by their relative lift. Architecturally, no single SAE training approach dominates; however, the top-ranked lifts are frequently occupied by *BatchTopK* and *Gated* variants, with *ReLU/JumpReLU* also contributing strongly and *TopK* showing more mixed outcomes. Overall, Δ *Token Confidence* yields consistent per-SAE gains, with variance decreasing and stability increasing as model size grows, while architectural diversity remains valuable for capturing the largest lifts.

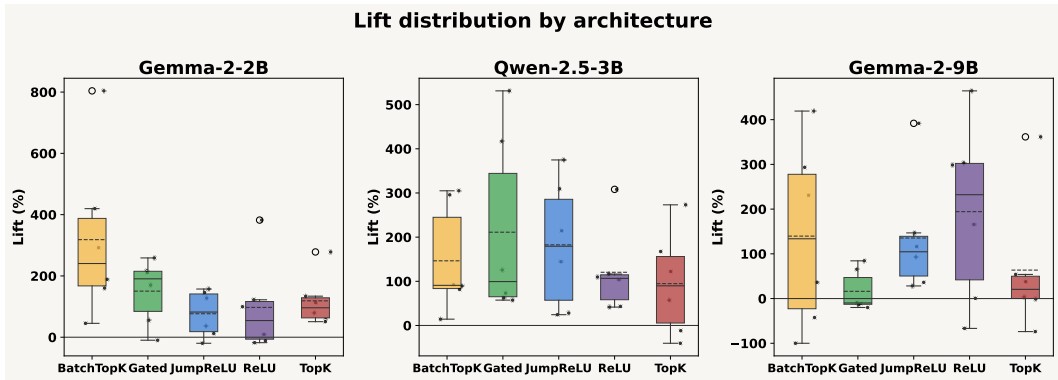

Figure 10: **Lift distributions by SAE architecture.** For each base model, box-and-whisker plots (with individual points overlaid) summarize the distribution of percentage lifts grouped by training architecture. Dashed horizontal lines denote the mean within each group, and whiskers follow the conventional interquartile rule.

Figure 10 groups lifts by architecture to visualize differences in central tendency and dispersion under the same selection and evaluation protocol. Read together with the per-SAE ranking, this distributional view helps disentangle architecture effects from model-specific variation and indicates which families tend to produce more stable or more variable outcomes after feature selection. *BatchTopK* and *Gated* generally occupy the highest central tendency with wide—but mostly posi-

tive—spread, especially on Gemma-2-2B and Qwen-2.5-3B. *BatchTopK* achieves the most stable and sizable *steering score*. Variance is largest for the smallest model (Gemma-2-2B), indicating architecture-sensitive wins at small scale.

## F    DATASET

**Training corpus for SAEs.** We train all SAEs on **The Common Pile v0.1** (Kandpal et al., 2025), an openly licensed ∼8 TB text collection built for LLM pretraining from ∼30 sources spanning research papers, code, books, encyclopedias, educational materials, and speech transcripts. The corpus was curated as a principled alternative to unlicensed web text and has been validated by training competitive 7B models on 1–2T tokens. We use it as the sole pretraining dataset for all SAE runs. More training details provided in Appendix A.2.

**CONCEPT100 for steering utility.** To evaluate steering, we construct CONCEPT100: a compact benchmark of *100 human-readable concept descriptions* per evaluation set, produced automatically by our interpretability pipeline (Appendix B). Each entry is a pair (`layer_feature_id, description`) that summarizes a latent's activation pattern in plain language (e.g., mathematical symbols, scientific terms, pronouns, or domain phrases). These descriptions are supplied to the AXBENCH judge when computing *steering score*. The examples below illustrate the style and domain coverage.

**Gemma-2-9B, BatchTopK, $L_0 \approx 80$.** Ten examples:

1. `20_14429`: concepts related to optical communication systems and their performance characteristics
2. `20_5795`: specific technical terms and chemical compounds often related to scientific contexts
3. `20_7908`: terms related to gravitational lensing and its effects in cosmology
4. `20_3042`: pronouns and verbs indicating relationships or contributions in various contexts
5. `20_11897`: scientific measurements and units related to energy, concentration, or biological data
6. `20_12944`: terms related to cell types and apoptosis mechanisms in scientific contexts
7. `20_8796`: references to specific authors and statistical concepts in mathematical contexts
8. `20_6430`: the phrase "action" in mathematical and theoretical contexts
9. `20_2220`: chemical elements and compounds, particularly including metals and metal-related terms
10. `20_585`: various forms of the word "energy" and related concepts in scientific contexts

**Qwen-2.5-3B, Gated, $L_0 \approx 72$.** Ten examples:

1. `17_15113`: terms related to mathematical concepts and various scientific names or terms
2. `17_11476`: the phrase "as a function of" in contexts of measurement and analysis
3. `17_162`: mathematical symbols and concepts related to coordinates, magnitudes, and parameters in equations
4. `17_2552`: dataset identifiers and technical terms common in research and academic documents
5. `17_16377`: mathematical notation and technical terms commonly found in formal documents
6. `17_3195`: demographic, clinical, and biological characteristics in study populations and related comparisons

7. `17_9186`: specific technical terms and concepts related to networking and programming

8. `17_11487`: mathematical notation and variables related to functions and equations

9. `17_14657`: mathematical notations and structures involving angle brackets and properties of functions

10. `17_1256`: terms related to errors and error correction in coding theory and quantum operations

**Gemma-2-2B, JumpReLU, $L_0 \approx 81$.** Ten examples:

1. `20_11531`: terms related to sports, programming, or specific keywords from various contexts

2. `20_10460`: terms related to fractional differential equations and numerical methods for solving them

3. `20_4882`: terms related to asymptotic theory, robustness, and statistical estimation methods

4. `20_4425`: first-person plural pronouns and expressions of intention or conjecture

5. `20_372`: technical terms related to measurement and structure in scientific contexts

6. `20_9999`: the word "from" and contexts implying deviation or distance from something

7. `20_9703`: the word "based" in various contexts of theoretical foundations and methodologies

8. `20_15509`: technical or numerical concepts in a variety of contexts

9. `20_8218`: phrases indicating conditions or assumptions that must be met in theoretical contexts

10. `20_4614`: time intervals and durations mentioned in the context of studies or observations

We currently maintain 90 SAEs (30 per base model). Beyond the CONCEPT100 sets evaluated in this paper, we have constructed the CONCEPT1000 and CONCEPT16K suites that scale the number of human-readable concepts up to 16K. We will extend training and evaluation to these larger suites in forthcoming releases to further substantiate the reliability and generality of this work.

## G  ANALYSIS OF STEERING STRENGTH ACROSS INTERPRETABILITY LEVELS

To directly address the concern that highly interpretable SAE features might have negligible steering effects, we conduct an additional analysis combining interpretability scores with steering scores at the feature level.

We do not find that interpretable SAE features have negligible steering effects. As the first set of panels in Figure 11 shows, high-interpretability features span the full range of steering strengths, including many strongly steering latents; high-interpretability features are not dominated by weak steering. Conversely, the second set of panels in Figure 11 shows that strongly steering features (top steering quartile) also cover almost the full range of interpretability scores, and many of them are only moderately or weakly interpretable according to interpretability score. In the SAEs under Qwen-2.5-3B, we found that features with extremely high steering scores also tend to have higher interpretability scores. This explains why, in Appendix J, only the SAE model under Qwen-2.5-3B shows improved steering performance when using high interpretability scores. However, below 1.0, the performance becomes highly dispersed again.

Taken together, these two views indicate that (i) high interpretability does not guarantee strong steering, and (ii) strong steering does not require high interpretability. In other words, feature-level interpretability scores provide limited information about how strongly a latent direction pushes the logits, reinforcing our claim that interpretability and steering strength are nearly orthogonal at the feature level.

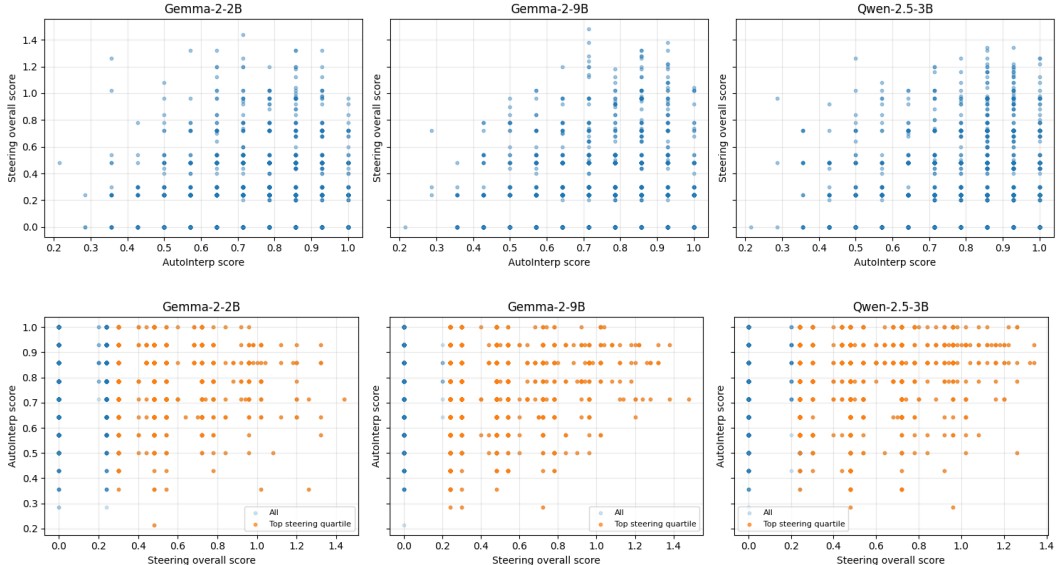

Figure 11: **Feature-level relationship between interpretability and steering strength across models.** Panels are shown separately for Gemma-2-2B, Gemma-2-9B, and Qwen-2.5-3B. The first set of panels plots interpretability score (AutoInterp) against steering overall score for every latent. The second set of panels flips the axes and highlights the top steering quartile in orange, with all latents shown in blue.

## H INSTRUCTION-FOLLOWING UNDER FEATURE-SELECTION STRATEGIES

To address whether our feature-selection metric surfaces features that make model outputs incoherent, we compare the AxBench *Instruction* subscore when steering with three strategies: using all SAE features ("SAE Base"), using features selected by the Output Score baseline, and using features selected by our $\Delta$ Token Confidence metric. For each model and SAE architecture, we aggregate the instruction subscores of the steered generations and plot them in Figure 12.

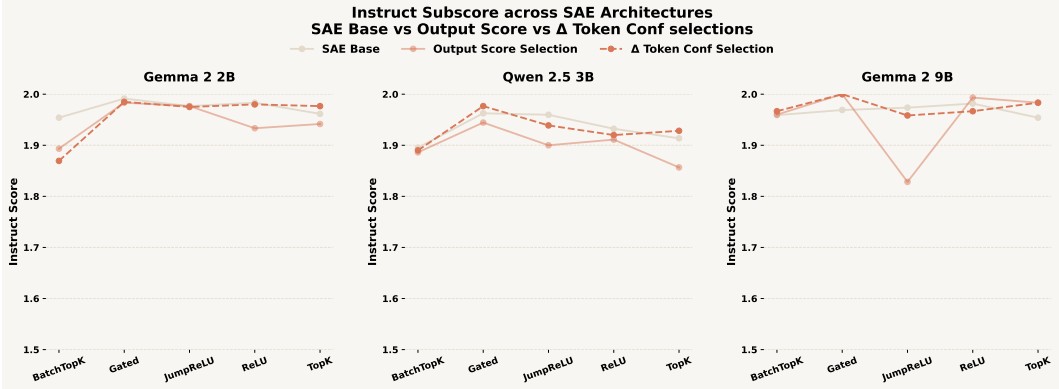

Figure 12: **Instruction subscore across SAE architectures when steering with different feature-selection strategies.** For each model, we compare the AxBench *Instruction* score under three setups: using all SAE features (SAE Base), using Output Score selection, and using $\Delta$ Token Confidence selection.

Across all models, both selection methods keep the instruction subscore close to the base level (typically in the 1.5–2.0 range on a 0–2 scale), indicating that neither Output Score nor $\Delta$ Token Confidence systematically degrades instruction following. This behavior is expected because coherence

is primarily controlled by the *steering factor* $\alpha$ in the intervention

$$x^{\text{steer}} = x + (\alpha m_f) \cdot v_f,$$

which scales how strongly a feature $f$ is injected into the hidden state. AxBench already tunes $\alpha$ per feature to balance concept shift against instruction following and fluency. The role of $\Delta$ Token Confidence is not to push the model into incoherent regimes, but to select features for which even small interventions induce a strong, directional change in the token-level logits.

## I EXPLORING THE STEERING UTILITY OF SAEs WITH HIGH INTERPRETABILITY

To directly test whether selecting features by interpretability also selects for high steering utility, we run a controlled comparison at the feature level. For each SAE in our three model–layer settings, we compute three steering configurations on AxBench: (i) **SAE Base**, which steers with all evaluated features; (ii) **Interpretability selection**, which steers using only the top-$p\%$ latents by AutoInterp score (we use $p = 10\%$ in Figure 13); (iii) $\Delta$ **Token Confidence selection**

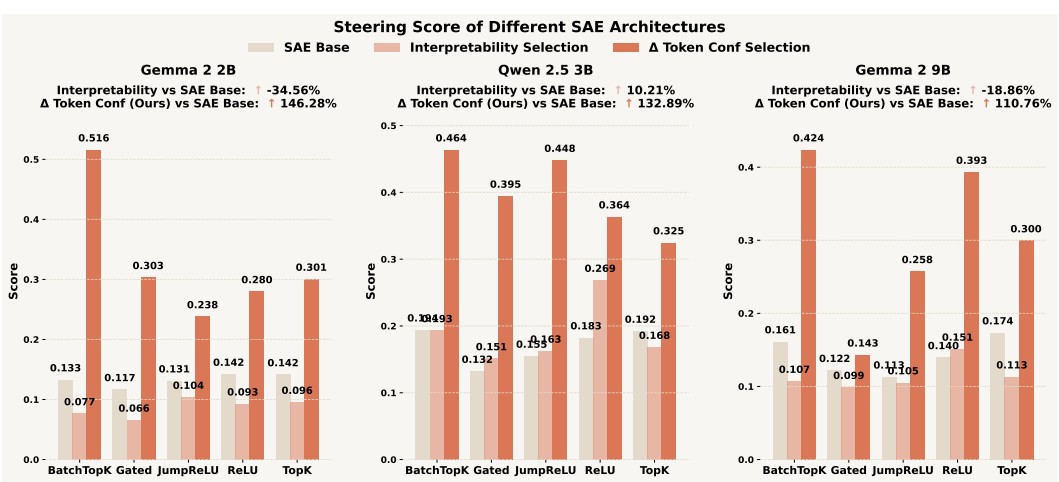

Figure 13: **Steering scores under different feature-selection strategies across SAE architectures and three models.** Bars show the AxBench overall steering score for (left) steering with all features (*SAE Base*), (middle) steering with the top-$10\%$ most interpretable features (AutoInterp), and (right) steering with features selected by $\Delta$ Token Confidence. The banners report the macro-average percentage lift over the SAE Base.

Across all three models, selecting features by interpretability alone yields at best modest and often negative changes in steering strength relative to using all SAE features (about 35% for Gemma-2-2B, +10% for Qwen-2.5-3B, and 19% for Gemma-2-9B), whereas Token Confidence consistently delivers large gains of roughly 1.1–1.5× over the same SAE base. This pattern indicates that feature-level interpretability is a weak signal for steering utility, while Token Confidence reliably concentrates probability mass on features that produce substantially stronger steering effects.

## J SIGNIFICANCE ANALYSIS FOR DIFFERENT SAE STEERING METHODS

To address the question of whether the improvements in Figure 4 are statistically significant, we reran the analysis at the SAE level and, for each model and architecture, computed the mean steering score and its 95% confidence interval across SAEs (shown as error bars in Figure 14). In addition, we performed paired bootstrap tests over SAEs for three contrasts: (i) $\Delta$ Token Confidence selection vs. SAE Base, and (ii) $\Delta$ Token Confidence selection vs. Output Score selection. Finally, we repeated the paired bootstrap test pooled over all 90 SAEs.

The error bars in Figure 14 show that $\Delta$ Token Confidence selection consistently lifts steering scores above the SAE Base across architectures and models.

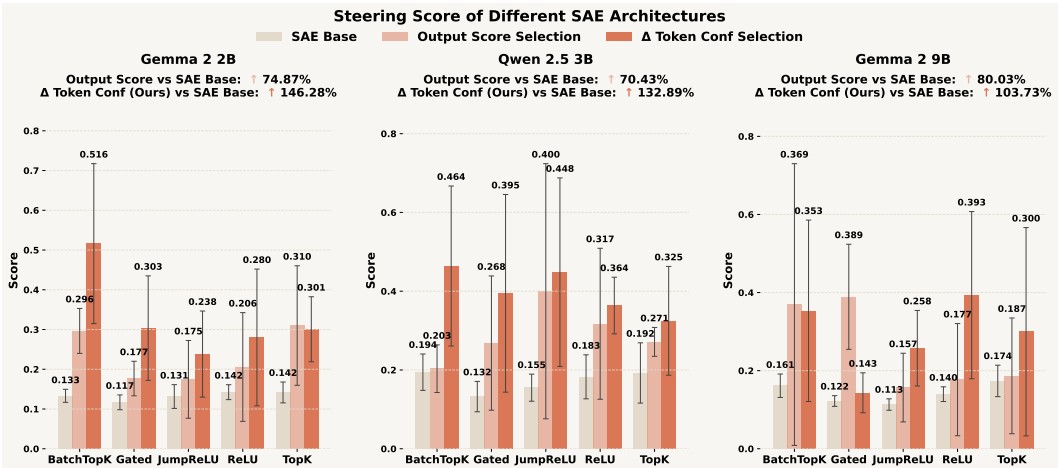

Figure 14: **Error-bar version of Figure 4.** Mean steering score and 95% confidence intervals across SAEs for the SAE Base, Output Score selection, and $\Delta$ Token Confidence selection, grouped by architecture and model. When aggregated across SAEs, there is a statistically significant difference between $\Delta$ Token Confidence and the baselines.

In this experiment, steering scores are first computed for each of the 90 SAEs under three conditions: SAE Base (no feature selection), Output Score-based selection, and $\Delta$TokenConf-based selection. We then form per-SAE paired differences and apply a paired bootstrap over SAEs to estimate the mean difference, its 95% confidence interval, and a two-sided $p$-value, yielding the global significance results summarized in Table 6.

Table 6: Pooled paired-bootstrap comparison of steering scores across all 90 SAEs. For each contrast, we report the mean per-SAE difference in steering score, a 95% bootstrap confidence interval, the two-sided $p$-value, and whether the effect is significant at the 5% level.

| Comparison | Mean difference | 95% bootstrap CI | Two-sided $p$ | Significant at 5%? |
|---|---|---|---|---|
| $\Delta$TokenConf $-$ Base | 0.1899 | [0.1454, 0.2339] | $\approx 0.0000$ | Yes |
| $\Delta$TokenConf $-$ Output Score | 0.0786 | [0.0261, 0.1305] | $\approx 0.0032$ | Yes |

Concretely, we find that our selection method yields a statistically significant improvement in steering score both relative to the SAE Base and relative to the Output Score selection.

## K ANALYSIS OF GAP BETWEEN SAE INTERPRETABILITY AND UTILITY

Regarding *why there is a gap*, our view is that the root cause lies in the training paradigm of SAEs. Current SAE training is almost entirely reconstruction-centric: the objective is to accurately reconstruct internal activations while enforcing sparsity, so as to obtain a more monosemantic basis in a higher-dimensional space. This is precisely what makes SAE features interpretable, but it does not directly optimize how useful those features are for steering. In other words, the training objective is aligned with *reconstruction fidelity and monosemanticity*, not with *control over model outputs*.

To make this concrete, we perform a pairwise Kendall–$\tau_b$ analysis over all 90 SAEs, relating interpretability, steering, and two standard reconstruction-side metrics (both higher-is-better): (i) the **CE loss score** and (ii) **explained variance**. Table 7 reports $\tau_b$, bootstrap standard error, a 95% bootstrap confidence interval, and a permutation $p$-value for the null $H_0 : \tau_b = 0$ for each pair of metrics.

As Table 7 shows, the reconstruction metrics exhibit only a *weak* association with steering ($|\tau_b| \approx 0.2$), but a substantially *stronger* association with interpretability ($|\tau_b| \approx 0.4$). In other words, the reconstruction-focused training objective is much more predictive of which SAEs score well under interpretability metrics than of which SAEs are actually good for steering, and is close to orthogonal

Table 7: Pairwise Kendall–$\tau_b$ between interpretability, reconstruction metrics, and steering score over all 90 SAEs. "CE loss score" and "explained variance" are standard post-hoc reconstruction metrics (higher is better).

| Pair | $\tau_b$ | SE | 95% bootstrap CI | perm $p$ ( $H_0 : \tau_b = 0$ ) |
|---|---|---|---|---|
| Interpretability – CE loss score | $-0.433$ | 0.056 | $[-0.539, -0.322]$ | 0.0002 |
| Interpretability – Explained variance | $-0.405$ | 0.059 | $[-0.521, -0.289]$ | 0.0002 |
| CE loss score – Steering | $-0.243$ | 0.067 | $[-0.377, -0.112]$ | 0.0006 |
| Explained variance – Steering | $-0.195$ | 0.066 | $[-0.323, -0.062]$ | 0.0062 |

to steering utility. This supports our claim that the observed interpretability–utility gap is not an artifact of our evaluation, but a structural consequence of the *current SAE training paradigm*, in which reconstruction fidelity—rather than steering behavior—is the primary optimization target.

## L  ABLATION ON THE TOP-$k$ IN $\Delta$ TOKEN CONFIDENCE

In our $\Delta$ *token confidence* selection, we measure how much a feature shifts the model's confidence over the top-$k$ tokens in the output distribution. Unless otherwise noted, all main experiments use $k = 1$, so that the score directly tracks the change in probability of the top-1 predicted token, which typically dominates next-token behavior in standard generation settings.

To verify that this choice is not artificially restrictive, we run an ablation over $k \in \{1, 3, 5, 10\}$ on Gemma-2-2B and Qwen-2.5-3B. Table 8 reports the resulting steering scores (higher is better) after applying $\Delta$ token confidence selection.

Table 8: Ablation over the top-$k$ parameter in $\Delta$ token confidence selection.

| $k$ (top-$k$) | Gemma-2-2B | Qwen-2.5-3B |
|---|---|---|
| 1 | 0.328 | 0.399 |
| 3 | 0.264 | 0.370 |
| 5 | 0.314 | 0.302 |
| 10 | 0.245 | 0.343 |

All tested values of $k$ yield substantial gains over the SAE baseline (not shown), and $k = 1$ is in fact the best-performing setting on both models. We therefore adopt $k = 1$ as an empirically validated default in all main experiments.

## M  EXAMPLES OF INTERPRETABLE-BUT-UNHELPFUL VS. HELPFUL-BUT-HARD-TO-EXPLAIN FEATURES

To complement our quantitative analysis, we provide qualitative examples of features where interpretability and steering utility diverge. Type A features are *highly interpretable but not steerable* (high interpretability score, near-zero steering score), while Type B features are *moderately interpretable but strongly steerable*. All generations in Table 9 are produced with temperature 0 using the fixed prefix ``I heard that'' and intervening on a single feature.

Table 9: Qualitative examples of features where interpretability and steering diverge. Type A: interpretable but not steerable (interpretability $\approx 1.0$, steering $\approx 0$). Type B: moderately interpretable but clearly steerable. All generations use temperature 0 with the prefix "I heard that".

| Field | Gemma 2 2B (A) | Qwen 2.5 3B (A) | Gemma 2 2B (B) | Qwen 2.5 3B (B) |
|---|---|---|---|---|
| Type | A | A | B | B |
| Latent id | 6168 | 12527 | 589 | 2392 |
| Architecture | BatchTopK, $L_0$=320 | BatchTopK, $L_0$=160 | ReLU, $L_0$=156 | JumpReLU, $L_0$=166 |
| Interpretability | 1.00 | 1.00 | 0.57 | 0.50 |
| Steering score | 0.00 | 0.00 | 1.32 | 1.02 |
| Explanation | the symbol 'pi' in mathematical expressions | terms related to differentiation in mathematical contexts | words related to actions or processes in various contexts | numerical values related to comparisons |
| After steering | I heard that there are a lot of people who live with their own thoughts in their heads. | I heard that it is Math and Science, did you hear that it is Math and Science? | I heard that The Body Shop **is planning to open** in Central World, I **used to go** to Bangna branch. | I heard that **a little bit larger than twice the actual, 2000 feet (2,500 meters)** worth of this, which is a mile of it, **three times** it's actual. |

