# OpenReview forum: "Does Higher Interpretability Imply Better Utility? A Pairwise Analysis on Sparse Autoencoders"
_ICLR.cc/2026/Conference — ICLR 2026 Poster_

### Official Review · Reviewer_vNrz · 2025-10-31

**Soundness:** 4
**Presentation:** 4
**Contribution:** 4
**Rating:** 8
**Confidence:** 4

**Summary:**

This work investigates the correlation of interpretability and steering utility of SAE latents by comparing rankings via  kendall's tau coefficient, and finds low correlation. After introducing a new method for selecting SAE features for steering, the correlation surprisingly vanishes.

**Strengths:**

- This paper provides quantitative evidence that interpretability scores are not indicative of steering utility. This contribution is relevant to the field -- interpretability is often treated as a universal score for the quality of learned features. I personally updated on putting less weight on automated interpretability as a measure for overall SAE quality.
- The writing is very clear, highlighting key takeaways, and motivating experiments.
- They train a wide range of SAEs
- Controlling for confounders when computing the correlation.
- The paper uses established metrics for interpretability (SAEBench) and steering success (AxBench)

**Weaknesses:**

- Looking at any code file in the anonymous repo link throws the error "The requested file is not found."

**Questions:**

- Should we use SAE latents for steering at all? How does the delta token confidence based steering compare to non-SAE based steering methods from AxBench?
- Are you open-sourcing the trained suite of SAEs?
- Can you provide max-activating inputs for features selected by delta-token-confidence? I'd be interested to see whether they are interpretable at all.
- Based on your results, do you deem interpretability scores as actively misleading, should we only rely on steering ability when judging SAE quality? Or are interpretability and steering utility rather complementary characteristics?

---

> ### Author Response · Authors · 2025-11-21
>
> We sincerely thank the reviewer for the insightful comments.
> >1. Looking at any code file in the anonymous repo link throws the error "The requested file is not found."
>
> Our experiments rely on large artifacts (SAE weights, feature lists, concept datasets) that could not be bundled in the anonymous submission. The camera-ready version will include a public repository with all code, trained SAEs, and scripts to download the necessary resources.
> >2. Should we use SAE latents for steering at all? How does...compare to non-SAE based steering methods from AxBench?
>
> Our main message is not that SAEs are useless for steering, but highlight a crucial research direction: effectively using SAEs for steering will likely require either **advanced post-training feature-selection protocols** or fundamentally **new SAE training paradigms that are explicitly utility-oriented**. We also add several other non-SAE-based steering methods to our experiments on three models to support our claim.
>
> |Method|Gemma-2-2B|Qwen-2.5-3B|Gemma-2-9B|
> |-|-|-|-|
> |Probe|0.071|0.094|0.080|
> |PCA|0.081|0.108|0.092|
> |LAT|0.088|0.113|0.098|
> |DiffMean|0.222|0.285|0.239|
> |SAE-based|0.133|0.171|0.142|
> |+Output|0.233|0.292|0.255|
> |**+ΔC_k (Ours)**|**0.328**|**0.399**|**0.289**|
>
> First, SAE-based steering intervenes directly in hidden representations without modifying model weights, enabling more lightweight control over model behavior compared with non-representation-based approaches (like fine-tuning).
>
> Second, within the family of representation-based steering methods, SAE-based steering has three key advantages:
> - Methods such as DiffMean and Probe require, constructing explicit positive and negative training sets, whereas SAE-based steering does not.
> - In our experiments, SAE-based steering achieves higher steering scores than representation-based methods such as PCA, LAT, and Probe, and after applying feature selection, it outperforms DiffMean.
> - Steering vectors derived from SAE features tend to be more semantically monosemanticity and thus more interpretable than the directions obtained from these alternative methods.
>
> We believe that SAE still has great potential in ensuring the controllability and safety of model output, and we hope to inspire a new training paradigm for SAEs by emphasizing the critical role of the interpretability–utility gap, rather than discouraging the use of SAEs for steering.
> >3. Are you open-sourcing the trained suite of SAEs?
>
> Yes. We plan to release the full suite of trained SAEs, together with code to load and use them for both interpretability and steering experiments.
> >4. Can you provide max-activating inputs for features selected by delta-token-confidence?
>
> We have inspected the max-activating inputs for the features selected by Δ token confidence.
> | Model|SAE|Feature_id|Explanation|Interp_score|Max-activating inputs (top3)|
> |-|-|-|-|-|-|
> |Gemma-2-2B|BatchtopK|10586| the concept of ‘agreement’ in various contexts|1|1. “to establish models, the **agreement is** very convincing…”2. “For the latter, good **agreement of**…”3. “chain length...revealed good **agreement with** results reported by…”|
> |Qwen-2.5-3B|BatchtopK|2374| mathematical symbols and expressions related to logarithms and convergence |0.9286|1.  “$\log(360\mu)$ appears in the transformed feature expression.”2. “$Q_{n,\xi} \to 0$ as $n\to\infty$ indicating convergence.”3.  “$K_{\phi}^{i\,\prime} \to K$ showing Jacobian convergence.”|
> |Gemma-2-9B|JumpReLU|6504| phrases indicating outcomes expressed by ‘result’ and ‘resulting’ in various contexts|1|1. “between psychological distress and health behaviors...can **result from** the former…”2.“This sub-optimal topology can **result in** unnecessary processing overhead…”3. “measures that disrupt the chain of transmission of GAS will **result in** a decreased burden…”|
>
> We find that many features are indeed interpretable and we will include these results in the appendix.
> >5. Do you deem interpretability scores as actively misleading...?
>
> Our results do not claim that interpretability scores are “misleading” in general, but they do show that current interpretability metrics are only a weak proxy for steering utility.
>
> SAEs are used for two main purposes:
> - Extract interpretable intermediate features for discovering domain-specific concepts [1], where human interpretability is crucial,
> - Steer model outputs to mitigate hallucinations [2] and bias [3], where we do not recommend using interpretability scores alone to assess SAE quality. In our experiments, interpretability and steering utility are weakly but positively correlated, so we view them as complementary. We plan to explore training paradigms that jointly optimize both.
>
> [1] Sparse autoencoders reveal selective remapping of visual concepts during adaptation ICLR 2025
>
> [2] Do I Know This Entity? Knowledge Awareness and Hallucinations in Language Models ICLR 2025
>
> [3] Evaluating Feature Steering: A Case Study in Mitigating Social Biases Anthropic Blog

---

> > ### Author Response · Authors · 2025-11-27
> > **Gentle Follow-Up on Rebuttal**
> >
> > Dear Reviewer vNrz,
> >
> > We are writing to kindly follow up and check whether you have any additional questions or comments, or if our responses have already addressed your concerns. We would be happy to continue the discussion during the open-review period, and we hope that our detailed clarifications help convey the quality and contributions of our work more compellingly.
> >
> > Thank you again for your time, consideration, and engagement.
> >
> > Sincerely,
> >
> > The Authors

---

### Official Review · Reviewer_2ZdN · 2025-11-01

**Soundness:** 2
**Presentation:** 3
**Contribution:** 2
**Rating:** 6
**Confidence:** 4

**Summary:**

This paper studies two aspects of SAEs evaluation: the interpretability aspect and the utility aspect (especially steering). The authors have conducted correlation analysis on 90 SAEs from 3 base models and find that there is little to no correlation between interpretability and utility scores. The authors also propose a new method that improves steering performance, and find the new method further weakens the correlation between interpretability and utility.

**Strengths:**

* This work offers an interesting perspective on SAE evaluation. In particular, the authors frame it as a tension between how interpretable the features are, i.e., interpretability score, and how useful the features are for steering, i.e., utility score. This framing could inspire new evaluation metrics and design on SAEs.
* The authors have conducted extensive experiment on 90 SAEs over 3 base models that cover common SAE architecture variations. This provides solid emperical evidence.

**Weaknesses:**

*  While the paper frames the findings as a tension between "interpretability" and "utility", the experiment operationalized "interpretability" and "utility" in a much narrower sense, i.e., as performance on SAEBench (the auto-interp task) and AxBench (the steering task).
    * For "Interpretability", it is unclear what exactly "interpretability" entails in the paper. The auto-interp score is not an intrinsic metric to the feature space learned by SAE, but rather confounded with the quality of the auto-interp pipeline and how closely it is aligned to human understandable concept space.
    * The "utility" is exclusively measured as "steering utility", while there are clearly other downstream applications of SAEs. For example, [Peng et al. 2025](https://arxiv.org/pdf/2506.23845) argues that SAE could be used on discovering unknown concepts. I would encourage the authors to be precise here and quantify the scope of study to match what actually done in the paper, for example, in the paper title, directly stating "steering utility" instead of "utility".

* The authors experimented with how different SAE architecture/features affect the gap, however, there is still a bit lack of insights on why there is a gap between "interpretability" and "utility". In particular, both interpretability score and utility score are extrinsic evaluation metrics that measure the success of a system as a whole, with SAE being one component. For example, interpretability score relies on the auto-interp pipeline, which is known to produce inaccurate description sometimes and these descriptions are often biased towards the role of the feature space in detecting concepts in inputs. Thus, is the gap between interpretability and utility a failure of the auto-interp pipeline, or more of a fundamental problem of SAEs?

**Questions:**

Please see the weakness. In particular, it would be great if authors could provide more insights/analyses into why there is a gap between interpretability and utility.

---

> ### Author Response · Authors · 2025-11-21
>
> Thank you for raising these valuable suggestions for improving our work!
> >1. For "Interpretability", it is unclear what exactly "interpretability" entails in the paper...The "utility" is exclusively measured as "steering utility"...directly stating "steering utility" instead of "utility".
>
> (1) First, our use of automated interpretability follows SAEBench [1], which explicitly treats LLM-based description-from-examples plus held-out prediction (Figure 1) as a measure of SAE interpretability. To address potential confounds from the quality or alignment of the auto-interp pipeline, we follow [2] to broaden the notion of SAE interpretability in different dimensions. We additionally report three complementary interpretability measures for each SAE:  (i) **Embedding**: latent–concept embedding similarity;  (ii) **Surprisal**: relationship between latent activations and surprisal;  (iii) **Fuzzing**: robustness of latents under prompt perturbations.
>
> | Pair|$\tau_b$|SE| 95% bootstrap CI |$p$ (H0: $\tau_b$ = 0)|
> |-|-|-|-|-|
> |Embedding–Steering|0.27|0.08|[0.12, 0.42]|0.0034|
> |Surprisal–Steering|0.22|0.08| [0.06, 0.37]|0.0128|
> |Fuzzing–Steering|0.26|0.08|[0.09, 0.40]|0.0050|
>
> We observe that across all correlational interpretability,  $\tau_b$ lies in the 0.2–0.3 range, which closely matches our **Key Observation 1**: Interpretability shows a relatively weak positive correlation with steering performance, highlighting a notable gap between interpretability and utility across SAEs. Therefore, we can eliminate interference from the quality of the auto-interp pipeline.
>
> (2) Discovering unknown concepts is more like a **property** of SAE itself (separating interpretable features in a high-dimensional space). What we want to express is the utility of SAE in downstream tasks (using features for steering). Meanwile, our contribution is a **general quantitative framework**. This framework is not tied to steering and can be applied to other downstream uses of SAEs. To reflect the scope more precisely, we will consider changing the title to ‘Steering Utility’ based on your suggestion and to clearly state that other downstream uses of SAEs in the related work section.
>
> [1] SAEBench: A Comprehensive Benchmark for Sparse Autoencoders in Language Model Interpretability ICML 2025
>
> [2] Automatically Interpreting Millions of Features in Large Language Models ICML2025
>
> >2. There is still a bit lack of insights on why there is a gap between "interpretability" and "utility"...is the gap between interpretability and utility a failure of the auto-interp pipeline, or more of a fundamental problem of SAEs?
>
> (1) Regarding 'why there is a gap', our view is that the root cause lies in the training method of SAE. The core objective is to reconstruct and obtain more monosemantic basis in a higher-dimensional space. This is one of the reasons why SAE's features can be explained. When we focus on improving SAE's ability to reconstruct the input activations, we overlook SAE's steering utility. To confirm this, we performed a pairwise analysis of the two core metrics (higher is better) for evaluating the reconstruction effect after SAE training: (i) **CE loss score** and (ii) **Explained Variance**, with steering score.
> | Pair|$\tau_b$|SE|95% bootstrap CI|perm p (H0: $\tau_b$ = 0)|
> |-|-|-|-|-|
> | Interpretability–CE loss score|-0.433| 0.056| [-0.539, -0.322]|0.0002|
> | Interpretability–Explained variance |-0.405|0.059|[-0.521, -0.289] |0.0002|
> | CE loss score–Steering|-0.243|0.067|[-0.377, -0.112]|0.0006|
> | Explained Variance–Steering|-0.195|0.066|[-0.323, -0.062]|0.0062|
>
> As shown in the table, The reconstruction metrics show only a **weak association with steering** ($|\tau_b|$ ≈ 0.2), but a **stronger association with interpretability** ($|\tau_b$| ≈ 0.4). In other words, the training objective is much more predictive of which SAEs look good under interpretability metrics than of which SAEs are good for steering, and is approximately orthogonal to steering utility. There is a mismatch between the training objective (reconstruction) and steering utility, so we think the interpretability–utility gap is fundamental problem **in the current training paradigm, where reconstruction fidelity—not steering—is the primary optimization target.**
>
> (2) To avoid the interpretability score relying solely on the auto-interp pipeline, we supplemented our answer to the first question with experiments. The results are **consistent with our insights and avoiding the problem of a failure of the auto-interp pipeline.** (Please refer to the first answer for details.)
>
> >3. Question: please see the weakness. In particular, it would be great if authors could provide more insights/analyses into why there is a gap between interpretability and utility.
>
> We agree with this concern and have expanded our analysis in the revised version (**Appendix K**). And we provided a detailed analysis in our second answer. Please refer to the analysis results in the second answer.

---

> > ### Author Response · Authors · 2025-11-27
> > **Gentle Follow-Up on Rebuttal**
> >
> > Dear Reviewer 2ZdN,
> >
> > We are writing to kindly follow up and check whether you have any additional questions or comments, or if our responses have already addressed your concerns. We would be happy to continue the discussion during the open-review period, and we hope that our detailed clarifications help convey the quality and contributions of our work more compellingly.
> >
> > Thank you again for your time, consideration, and engagement.
> >
> > Sincerely,
> >
> > The Authors

---

### Official Review · Reviewer_MCfE · 2025-11-01

**Soundness:** 2
**Presentation:** 3
**Contribution:** 2
**Rating:** 2
**Confidence:** 3

**Summary:**

This paper considers sparse autoencoders (SAEs), a popular technique in mechanistic interpretability, and seeks to uncover the relationship between the interpretability of a latent (i.e. to what extent the latent corresponds to a robust concept) and the utility of the latent (its ability to steer downstream generation towards or away from that concept). The paper uses two popular benchmarks to encode these ideas: SAEBench for interpretability and AxBench for utility. Despite the conventional wisdom being that interpretability and utility would be strongly correlated, the paper finds only a weak positive association (0.298). Furthermore, after filtering to only the top latents in terms of their impact on the logit distribution, the association drop to near 0 (whereas one might expect it to be positive). This is a counterintuitive result. The study is comprehensive across 90 SAEs from different architectures, sparcities etc.

**Strengths:**

- The study is comprehensive. It considers 90 SAEs with varying architectures and sparsity levels. This is a strong point of the paper since the results are generalisable to all SAEs. The one axis that isn't varied is the dictionary size, always being 16k; however, there is sufficient variation elsewhere.
- The paper is generally well-written and easy to read. Figures 1 and 2 are informative.
- The methodology used allows a hierarchical analysis of differences between models of different architectures, etc. I have questions about the statistical significance of these results (see next section), but this analysis is really nice since it allows claims of which architecture provides the smallest interpretability-utility gap.
- Fundamentally, this is an unexpected result and should be of interest to the community. It is odd that latents which are more interpretable are not necessarily more useful for steering!

**Weaknesses:**

I've ordered these by importance.
- **The definition of interpretability:** My main concern is around the definition of interpretability of a latent. The paper assumes a *correlational scoring* approach following one of the methods set out in [1]. Quoting from the paper under review, "the judge drafts the description from examples and then predicts, on a held-out set" (L135). This is correlational because it only considers when the latent activates and when it doesn't. The paper writes like this is the default definition of interpretability, and then compares the derived scores to the utility scores. However, in reality, people often evaluate interpretability using an *intervention scoring* method as well as just the correlational method [1]. To quote from Paulo et al. 2025 (the same paper used to define the AutoInterp metric:
> "intervention scoring, evaluates the interpretability of the effects of intervening on a feature, which we find explains features that are not recalled by existing methods."

    In short, the authors' operationalisation of *utility* is often used by other papers as a way of measuring *interpretability*. As a result, the distinction of interpretability/utiliy is not as obvious as it may seem. It remains a very interesting result that interpretability from correlational scoring seems to be unrelated to interpretability from interventional scoring, or utility, but the phrasing of the result needs a bit of work in my opinion.

- **Statistical testing:** Many of the results rely on the point value of the $\tau_b$; however, this would benefit from showing standard errors and doing hypothesis tests to examine whether it is statistically significantly different from 0. In particular, (i) when we change it to only consider those with high Token Confidence, is the change in $\tau_b$ statistically significant? (ii) When you compare different architectures, are any of those changes statistically significant? (iii) In Figure 4, if we add error bars to the plot, are any of the comparisons statistically significant? This seems really important to support the main results. Table 3 does show standard errors for the axis-controlled summaries, but the CIs are quite large, which makes me suspect that some of the results reported (e.g. the comparisons between architectures) are not significant.
- **Clarification on why the gap matters:** Most readers probably work it out quickly, but explicitly describing why the gap between interpretability and utility matters for downstream tasks would be nice. (In fact, explicitly defining "the gap" would also be helpful).
- **Examples:** As mentioned, I think this main result is weird and unexpected, which makes it interesting! It would be great to see some examples of features that are interpretable but not steerability, and vice versa.
- **How do you select $k$?** A smaller point, but how do you establish $k$ in the token confidence calculation? In the appendix, it says $k=1$ for all experiments. Is this the correct interpretation? If so, I feel this should be in the main text, and it feels somewhat limiting given that the metric is explained in the context of changing the logit distribution.
- **Writing:** In general, the writing is strong, but there are a few places where metrics are not clearly defined, e.g. *CONCEPT100*. The paper sends the reader to the appendix for a definition, but it would be best to define these terms in the main body.

This paper studies an interesting question and has a lot of potential, but I feel there are some critical questions that need to be addressed before it is ready for publication.

**Typos:**
- L053 missing year in citation.
- L212 citation capitalisation

---
[1] Paulo et al., 2025

**Questions:**

- I was confused by *steering gain* (sec. 4.3). From my understanding, this is a different metric to *steering score* in the earlier parts of the paper. If this is correct, are we calculating the same metric $tau_b$ in the section case (filtering for only the top token confidence latents)? If not, are these two $tau_b$ metrics directly comparable? The way I interpreted the method from the description is that we would filter for the top tokens by confidence latents, and then calculate the exact same metric, but I don't think this is the case, right? Thanks.

---

> ### Author Response · Authors · 2025-11-21
> **Detailed answers and experiments for the first question.**
>
> We thank the reviewers for raising important points that improved our work!
> >1. The definition of interpretability: My main concern is around the definition of interpretability of a latent...As a result, the distinction of interpretability/utiliy is not as obvious as it may seem.
>
> Thank you for raising this conceptual point about the definition of interpretability.
>
> First, our use of automated interpretability follows SAEBench [1], which explicitly treats LLM-based description-from-examples plus held-out prediction (Figure 1 in SAEBench) as a measure of SAE interpretability. Our work builds on this established formulation of the concept rather than introducing a new definition.
>
> To address your concern and align with Paulo et al. [2], we have now explicitly evaluated both correlational and interventional notions of interpretability in 90 SAEs. Following [2], we add four metrics in different dimensions of SAE interpretability (higher is better): (i) **Embedding similarity**, (ii) **Surprisal**, (iii) **Fuzzing**, and (iv) **Intervention scoring**. We then compute Kendall’s  $\tau_b$ between each interpretability metric and steering score.
>
> | Pair|$\tau_b$|SE| 95% bootstrap CI |$p$ (H0:$\tau_b$ = 0)|
> |-|-|-|-|-|
> |Embedding–Steering|0.27|0.08|[0.12, 0.42]|0.0034|
> |Surprisal–Steering|0.22|0.08| [0.06, 0.37]|0.0128|
> |Fuzzing–Steering|0.26|0.08|[0.09, 0.40]|0.0050|
> |Intervention–Steering|-0.33|0.08|[-0.49, -0.16]|0.0004|
>
> We observe:
>
> - Across all correlational interpretability,  $\tau_b$ lies in the 0.2–0.3 range, which **closely matches our Key Observation 1**: Interpretability shows a relatively weak positive correlation with steering performance, highlighting a notable gap between interpretability and utility across SAEs.
>
> - Intervention vs. Steering:  $\tau_b$ = −0.33. Intervention score is high for features whose interventions, at fixed KL strength, strongly and consistently push model outputs toward a single, easily described pattern, without considering instruction-following or fluency, whereas AxBench’s steering score is the harmonic mean of concept, instruction-following, and fluency judgments and therefore heavily penalizes exactly such aggressive, task-breaking interventions.
>
> This mirrors Paulo et al.’s own finding that their context-based correlational score and intervention scoring are negatively correlated for some layers, indicating that some features are best explained via their downstream effects rather than by where they activate. From table we can see: **features whose output effects are most “interpretable” in Paulo et al.’s sense are not the ones that maximise downstream steering utility.**
>
> Given this, we believe it is conceptually clearer to treat steering score as a utility metric that is largely orthogonal to interpretability (correlational *or* interventional), rather than as yet another interpretability score.
>
> [1] SAEBench: A Comprehensive Benchmark for Sparse Autoencoders in Language Model Interpretability ICML2025
>
> [2] Automatically Interpreting Millions of Features in Large Language Models ICML 2025

---

> > ### Author Response · Authors · 2025-11-21
> > **Detailed answers and experiments for the second question.**
> >
> > >2. Statistical testing: Many of the results rely on the point value...(i) when we change it to only consider those with high Token Confidence...(ii) When you compare different architectures...(iii) In Figure 4, if we add error bars to the plot...are not significant...Table 3 does show standard errors for the axis-controlled summaries
> >
> > (1) In response to (i), we have now explicitly tested whether the change in Kendall’s correlation is statistically significant when moving from the baseline steering score to the high $\Delta$ Token Confidence steering score.
> >
> > For all 90 SAEs in Table 1, the baseline association between interpretability and steering is $\tau_b = 0.2979$, indicating a clear positive correlation. After applying high $\Delta$ Token Confidence feature selection, the association drops to $\tau_b = 0.0823$ (Table 3), which is statistically indistinguishable from zero.
> >
> > To quantify this change, we directly test the null hypothesis:
> >  $H_0: \tau_b^{\text{before}} = \tau_b^{\text{after}}$
> >  by examining the paired difference $\Delta \tau_b = \tau_b^{\text{after}} - \tau_b^{\text{before}}$ using a bootstrap over SAEs:
> >
> > |Setting|$\tau_b$|Standard Error(SE)|95% bootstrap CI|Hypothesis test & p-value|Significance?|
> > |-|-|-|-|-|-|
> > |$\Delta=\tau_{\text{high}}-\tau_{\text{base}}$|$-0.2156$|$0.0762$|$[-0.3587,-0.0650]$|$H_0!:\ \Delta=0\rightarrow$ $p\approx0.0132$|Yes|
> >
> > Crucially, the paired difference $\Delta \tau_b = -0.2156$ has a 95% bootstrap confidence interval of $[-0.3587, -0.0650]$ and  $p \approx 0.0132$ (＜0.05), so the decrease in $\tau_b$ is statistically significant. This directly addresses (i): **the correlation between interpretability and steering utility significantly weakens after restricting to high $\Delta$ Token Confidence features and becomes statistically compatible with zero.**
> >
> > (2) Comparing different architectures is not our primary objective. Our goal is to explore the gap between interpretability and utility at the SAE level, and then consider whether a new training paradigm is needed to jointly optimize interpretability and utility. Here, we tested the comparisons highlighted in the main text (such as BatchTopK vs. Gated). This is a **framework** that can be used to compare different architectures (here shows a few examples). We directly test $H_0:\tau_b(\text{arch A}) = \tau_b(\text{arch B})$ using a two-sided paired bootstrap over SAEs.
> >
> > |Comparison(A−B)|Setting|$\Delta \tau_b=\tau_b(A)-\tau_b(B)$|95%CI|$p$|Significance?|
> > |-|-|-|-|-|-|
> > |BatchTopK−Gated|$g_{\text{base}}$|0.5229|$[0.0276,1.0010]$|0.0368|**Yes**|
> > |Gated−JumpReLU|$g_{\text{base}}$|-0.6275|$[-1.0909,-0.0976]$|0.0228|**Yes**|
> > |Gated−ReLU|$g_{\text{base}}$|-0.5621|$[-1.0231,-0.0544]$|0.0284|**Yes**|
> > |Gated−TopK|$g_{\text{base}}$|-0.5882|$[-1.0483,-0.0672]$|0.0272|**Yes**|
> > |ReLU−TopK|$g_{\text{high}}$|-0.5709|$[-1.0793,-0.0553]$|0.0348|**Yes**|
> >
> > Overall, these tests show that (1) the positive interpretability–steering association is **not driven by a single SAE architecture**, it is significant for multiple architectures and (2) architecture choice does matter: Gated exhibits reliably weaker correlation than the stronger-performing architectures.
> >
> > (3) We have added error bars to the plot and use a paired bootstrap over all 90 SAEs to support the insight: $\Delta$ Token Confidence reliably selects high-utility SAE features across models in **Appendix J (Figure 14)**. We then form per-SAE paired differences and apply a paired bootstrap over SAEs to estimate the mean difference, its confidence interval, and a two-sided p-value.
> > |Comparison|Mean difference|95% bootstrap CI|$p$|Significance?|
> > |-|-|-|-|-|
> > |ΔTokenConf−Base|0.1899|[0.1454,0.2339]|≈0.0000|Yes|
> > |ΔTokenConf−Output Score|0.0786|[0.0261,0.1305]|≈0.0032|Yes|
> >
> > Concretely, we find that our selection method yields a **statistically significant improvement** in steering score both relative to the SAE Base and relative to the Output Score selection.
> >
> > (4) In **Table 3 (updated)**, only a few subgroups are statistically significant ($p < 0.05$), indicating that after feature selection we do not observe a stable positive or negative correlation between interpretability and utility. The lack of significance does not affect our key observations; on the contrary, it validates our conclusion: after feature selection, we cannot verify the relationship between the interpretability and utility of SAEs, and the gap is deepened. Our focus here is precisely on the subset of high-utility SAEs selected by $\Delta$ Token Confidence, where the interpretability–utility association becomes small and statistically indistinguishable from zero, rather than on detailed comparisons between architectures. In this sense, the statistical testing in Table 3 supports our claim that **the interpretability–utility gap widens among high-utility features**.

---

> > > ### Author Response · Authors · 2025-11-21
> > > **Detailed answers and experiments for questions three, four, five, six, and seven.**
> > >
> > > > 3. Clarification on why the gap matters...describing why the gap between interpretability and utility matters for downstream tasks would be nice.
> > >
> > > - **Need for a new training paradigm in SAE.** First, the most practical problem is that researchers often do not know how well a pre-trained SAE performs in terms of steering. To truly evaluate utility, it is usually necessary to rely on LLM judges to score, which is both expensive and time-consuming. Therefore, a very natural question arises: can the interpretability score of an SAE be used as a priori indicator of its utility? While our paper highlights there is a gap, it will drive a new SAE training paradigm that emphasizes both reconstruction and utility.
> > > - **Over-reliance on SAE's interpretability metric.** Furthermore, many works have directly used pre-trained SAEs for steering in downstream tasks. Previous studies often implicitly assume that "as long as the SAE has good interpretability, it can be safely used for steering", ignoring the potential gap between the SAE's interpretability and utility.
> > >
> > > In summary, discussing this gap is crucial not only for SAE training but also for steering based on it. We will elaborate on this point more clearly and explicitly define "the gap"  in the revised version.
> > > >4. It would be great to see some examples of features that are interpretable but not steerable **(A)**, and vice versa **(B)**.
> > >
> > > We extract two examples from Gemma and Qwen respectively to demonstrate. Prefix is: I heard that.
> > > |Field|Gemma 2 2B|Qwen 2.5 3B|Gemma 2 2B|Qwen 2.5 3B|
> > > |-|-|-|-|-|
> > > |Type|A|A|B|B|
> > > |Latent id|6168|12527|589|2392|
> > > |Architecture|BatchTopK,L0=320|BatchTopK,L0=160|ReLU,L0=156|JumpReLU,L0=166|
> > > |Interpretability|1.00|1.00|0.57|0.50|
> > > |Steering|0.00|0.00|1.32|1.02|
> > > |Explanation|the symbol 'pi' in mathematical expressions|terms related to differentiation in mathematical contexts|words related to actions or processes in various contexts|numerical values related to comparisons|
> > > |after steer|I heard that there are a lot of people who live with their own thoughts in their heads.|I heard that it is Math and Science, did you hear that it is Math and Science?|I heard that The Body Shop **is planning to open** in Central World, I **used to go** to Bangna branch.|I heard that **a little bit larger than twice the actual**, **2000 feet (2,500 meters)** worth of this, which is a mile of it, **three times** it's actual.|
> > > >5. How do you select `k`? A smaller point, but how do you establish `k` in the token confidence calculation?
> > >
> > > Yes, we use $k = 1$ for all main experiments, and we will state this explicitly in the main text. We chose $k = 1$ because it directly tracks how much a feature shifts the probability of the top-1 predicted token. In common generation settings, the behavior of the next token is usually dominated by top-1 logit, so $C_1(p) = -\log p^{(1)}$ captures the most critical distribution changes.
> > > |top-k|Gemma-2-2B|Qwen-2.5-3B|
> > > |-|-|-|
> > > |1|0.328|0.399|
> > > |3|0.264|0.370|
> > > |5|0.314|0.302|
> > > |10|0.245|0.343|
> > >
> > > We have run ablation experiments before over $k \in \{1, 3, 5, 10\}$ on Gemma-2-2B and Qwen-2.5-3B (see table above): all values of $k$ yield large improvements over the SAE baseline, and $k = 1$ is in fact the best-performing setting on both models, so we treat it as an empirically validated default. We add experiments into **Appendix L**.
> > > >6. There are a few places where metrics are not clearly defined, e.g. `CONCEPT100`.
> > >
> > > In the revised version, we have added clearer definitions and descriptions of the metrics (including `CONCEPT100`) directly in the main body of the paper (**Section3.1**), rather than only in the appendix. We have also corrected the reported typos.
> > > >7. I was confused by *steering gain* (sec. 4.3)...are these two ` $\tau_b$` metrics directly comparable...
> > >
> > > We thank the reviewer for pointing this out and apologize for the confusion caused by our definition. The previous draft mixed two different notions (“steering score” vs. “steering gain”), which arose from a mix-up between draft versions in the submitted manuscript.
> > >
> > > In the revised version, we unify “utility” to mean the steering score itself rather than a relative gain. Table 3 now reports $\tau_b\big(\mu(\theta), g_{\text{high}}(\theta)\big)$, where $\mu(\theta)$ is the interpretability score and $g_{\text{high}}(\theta)$ is the steering score recomputed after selecting high Δ Token Confidence features for each SAE. This is directly comparable to Table 1, which reports $\tau_b\big(\mu(\theta), g_{\text{base}}(\theta)\big)$ using the original steering score $g_{\text{base}}(\theta)$.
> > >
> > > In the updated Table 3, the measured value is **$\tau_b\big(\mu(\theta), g_{\text{high}}(\theta)\big) = 0.0823$** which which remains consistent with our **key observation 3**: the interpretability-utility gap widens among high-utility features. To quantify this change, we have already tested the null hypothesis in our answer to the second question (please refer to the answer above).

---

> > > > ### Author Response · Authors · 2025-11-27
> > > > **Gentle Follow-Up on Rebuttal**
> > > >
> > > > Dear Reviewer MCfE,
> > > >
> > > > We are writing to kindly follow up and check whether you have any additional questions or comments, or if our responses have already addressed your concerns. We would be happy to continue the discussion during the open-review period, and we hope that our detailed clarifications help convey the quality and contributions of our work more compellingly.
> > > >
> > > > Thank you again for your time, consideration, and engagement!
> > > >
> > > > Sincerely,
> > > >
> > > > The Authors

---

> > > > > ### Comment · Reviewer_MCfE · 2025-11-27
> > > > >
> > > > > **1.** Sure, I appreciate that the community has used the SAEBench definition before and there is general use of this metric. Thanks for doing the additional experiments with the other metrics. I'm a bit confused why $\tau_b$ is negative for the interventional interpretability metric. I appreciate the comment about the AxBench steering score penalising aggressive steering, but would we not expect a strongly positive result, given the prior concern was that utility was essentially capturing something very similar to interventional interpretability? Paulo et al.'s finding that correlational score and interventional score are sometimes negatively correlated (i.e. some features can only be explained by their downstream effects), seems like a very similar finding to the $\tau_b$ between correlational score and steering score being weak. Would you mind conceptually explaining why interventional score and utility capture different properties?
> > > > >
> > > > > **2.** (i) Thanks for doing that. This analysis is clear and I think it would be worth adding a line to the paper. (ii) Sure, thanks for doing this. I know this is not the main claim of your paper but it is good that the sentence ``ReLU-like variants reinforcing the trend and Gated weakening it'' can be backed up statistically. (iii) Thanks for adding this figure. The error bars are fairly large, but I get that when aggregated, there is a statistically significant difference. (iv) Sure, this makes sense. Thanks for clarifying that.
> > > > >
> > > > > **3.** Thanks for explaining both of these points and committing to updating the definition a bit more clearly in the paper. I think most readers will be able to work this out, but it would help if it were spelt out a bit more.
> > > > >
> > > > > **4.** This is really interesting to see! I think examples like this would really help the paper if included. Perhaps also with temperature 0 behaviour pre-steering (or a representative example of the pre-steering behaviour).
> > > > >
> > > > > **5.** Thanks for adding these experiments. It would definitely be good to make it clear in the main text that $k=1$, as this was not my initial assumption when reading it.
> > > > >
> > > > > **6.** Thanks.
> > > > >
> > > > > **7.** Thanks for checking this and changing the paper. That makes much more sense now.

---

> > > > > > ### Author Response · Authors · 2025-11-28
> > > > > > **Detailed answers for the first question.**
> > > > > >
> > > > > > We sincerely thank the reviewer for the thoughtful follow-up comments and for the consideration of our responses!
> > > > > > >1: conceptually explaining why ... capture different properties?
> > > > > >
> > > > > > To clarify this, we argue that their objectives are misaligned, and do experiments for revealing an inverted-U relationship between them.
> > > > > >
> > > > > > **(1) The two metrics have different objectives**
> > > > > >
> > > > > > In Paulo et al., the interventional score for a feature with explanation $z$ is
> > > > > > $$ S = \\mathbb{E}\_x \\Big[ \\mathbb{E}\_{i \\sim G\_I(x)}[\\log p\_M(z\\mid i)] - \\mathbb{E}\_{g \\sim G(x)}[\\log p\_M(z\\mid g)] \\Big], $$
> > > > > > where $G(x)$ and $G_I(x)$ denote the base and intervened **Generator (G)** (i.e., $G(x)$ produces the base model outputs for prompt $x$, and $G_I(x)$ produces outputs under an **Intervention (I)** on the activation for feature $z$ while keeping $x$ fixed), and $p_M(z\mid \cdot)$ is the judge model. This metric ignores whether the original instruction is preserved or the text remains fluent; it only measures how strong and easy-to-describe the downstream effect is.
> > > > > >
> > > > > > While steering score is the harmonic mean of three LLM-judged components:
> > > > > > - Concept — target concept expressed
> > > > > > - Instruction — original instruction followed
> > > > > > - Fluency — output natural and coherent
> > > > > >
> > > > > > A feature only achieves a high steering score if it **simultaneously** (i) makes the concept salient, (ii) respects the instruction, and (iii) keeps the output fluent. As a result, steering score **heavily penalises “hard switches”** whose activation overwhelms the prompt, causes the model to ignore instructions, or produces repetitive text.
> > > > > >
> > > > > > So, informally:
> > > > > > - Interventional score ≈ *“how strongly and stably does this feature force the output into one pattern?”*
> > > > > > - Steering score ≈ *“how well does this feature inject the concept without breaking the task or fluency?”*
> > > > > >
> > > > > > **(2) This mismatch induces an inverted-U between intervention score and steering score**
> > > > > >
> > > > > > Conceptually, we expect three regimes:
> > > > > >
> > > > > > - **Small interventions**: the concept is weak, so steering score is low.
> > > > > > - **Moderate interventions**: the concept becomes clear while Instruction/Fluency are still mostly intact—this is where **steering score is maximal**.
> > > > > > - **Very strong interventions**: the feature behaves like a **hard switch**, which is ideal for interventional scoring, but it **over-steers** the model and harms Instruction or Fluency.
> > > > > >
> > > > > > When we bucket features by their interventional score into 4 quantiles, we obtain:
> > > > > > |Bin|Mean Interv|Mean Steer|
> > > > > > |-|-|-|
> > > > > > |1(20–40%)|0.3778|0.1453|
> > > > > > |2(40–60%)|0.4880|0.1567|
> > > > > > |3(60–80%)|0.5974|0.1536|
> > > > > > |4(80–100%)|0.7657|0.1307|
> > > > > >
> > > > > > From table, steering score peaks in the *moderate* regime (bin 2) and is lowest in the highest-intervention bin 4. To further check where high-utility features live, we look at the top 20% features by steering score and examine their interventional scores. These features are concentrated in bin 2, with:
> > > > > > - mean SteeringScore ≈ 0.41.
> > > > > > - Meanwhile, mean InterventionScore ≈ 0.47,
> > > > > >
> > > > > > So utility is maximised at moderate intervention strength, and decreases sharply once interventions become “hard switches”.
> > > > > >
> > > > > > **(3) The impact of this inverted-U on $\\tau\_b$**
> > > > > >
> > > > > > We compute Kendall’s $\\tau\_b$ between $\\{\\mu(\\theta\_i)\\}$ (intervention) and $\\{g(\\theta\_i)\\}$ (steering) exactly as in our paper:
> > > > > > $$ v\_{ij} = \\operatorname{sign}\\Big((\\mu(\\theta\_i)-\\mu(\\theta\_j))\\,(g(\\theta\_i)-g(\\theta\_j))\\Big)\\in\\{-1,0,+1\\}. $$
> > > > > > Our empirical inverted-U relationship between interventional score and steering score implies that, once interventions become “too strong”, increasing $\\mu(\\theta)$ typically decreases $g(\\theta)$. Consequently, for many SAE pairs $(\\theta\_i,\\theta\_j)$ we observe
> > > > > > $$ \\mu(\\theta\_i) > \\mu(\\theta\_j) \\quad\\Longrightarrow\\quad g(\\theta\_i) < g(\\theta\_j), $$
> > > > > > so that
> > > > > > $$ (\\mu(\\theta\_i)-\\mu(\\theta\_j))\\,(g(\\theta\_i)-g(\\theta\_j)) < 0 \\quad\\Rightarrow\\quad v\_{ij} = -1. $$
> > > > > >
> > > > > > Concretely, this is exactly what happens when comparing SAEs whose features concentrate in the strongest-intervention bin 4 (with higher $\mu(\theta)$ but lower steering score) against those concentrated in the moderate bin 2 (with lower $\mu(\theta)$ but higher steering score): their rankings by interventional score and by steering score are systematically reversed, contributing many $v_{ij} = -1$ terms and hence a negative $\tau_b$.
> > > > > >
> > > > > > **Increasing interventional score by forcing features into the hard-switch regime places them on the right side of the inverted-U,** where steering score is lower—this is exactly what the negative $\\tau\_b$ is capturing.
> > > > > >
> > > > > > Our findings thus extend Paulo et al.’s observation that different interpretability notions (correlational vs interventional) can disagree: even when both metrics are defined in terms of downstream outputs, **“features whose effects are easiest to describe” and “features that are actually best for controlled steering under task and fluency constraints” need not coincide.**

---

> > > > > > > ### Author Response · Authors · 2025-11-28
> > > > > > > **Replies to the remaining questions**
> > > > > > >
> > > > > > > >**2:** I think it would be worth adding a line to the paper...when aggregated, there is a statistically significant difference
> > > > > > >
> > > > > > > (i) We add a concise sentence in the main text explicitly stating this result and its statistical significance.
> > > > > > > (ii) We are glad this was helpful, and we will keep the sentence about ReLU-like variants and Gated SAEs, backed by the statistical tests, in the revised version. We ensure that all similar claims in the paper are supported by statistical significance tests.
> > > > > > > (iii) Your understanding is right: significance testing requires sufficient sample size, and at present each subgroup contains relatively few SAEs, so we obtain a statistically significant improvement when aggregating across subgroups. We will keep the figure with error bars and make the “aggregated, statistically significant improvement” point explicit in the caption and text. In addition, we plan to train more SAEs to further reduce these error bars.
> > > > > > > (iv) We are happy that this clarification resolves the concern, and we reflect it clearly in the revised draft.
> > > > > > >
> > > > > > > >**3:** It would help if it were spelt out a bit more.
> > > > > > >
> > > > > > > Thank you for this suggestion. In the revised version, we now explicitly spell out in the **Introduction** section why the interpretability–utility gap matters, briefly explaining when interpretability scores can be (mis)used as a proxy for utility and how this motivates SAE training objectives that jointly consider reconstruction, interpretability, and downstream steering performance.
> > > > > > >
> > > > > > > >**4:** Perhaps also with temperature 0 behaviour pre-steering.
> > > > > > >
> > > > > > > We appreciate this suggestion. In the revised version we add these qualitative examples in **Appendix M**, and explicitly state that all generations (pre- and post-steering, with the same “I heard that” prefix) are produced with temperature~0 (that's what we did before), so they reflect representative temperature-0 behaviour.
> > > > > > >
> > > > > > > Once again, we sincerely appreciate your careful consideration of each of our responses. We hope that our latest clarifications fully address your concerns, and we are grateful for your suggestions, which have helped us strengthen the paper; if any further questions remain, we would be very happy to address them.

---

### Official Review · Reviewer_AA1u · 2025-11-02

**Soundness:** 2
**Presentation:** 2
**Contribution:** 2
**Rating:** 2
**Confidence:** 3

**Summary:**

This paper studies the extent to which SAE interpretability metrics correlate with how effective an SAE's latents are for steering, finding a small positive association. They then introduce a metric—a feature's steering effect on the log probability of the most probable tokens—that selects for features that are very effective for steering.

**Strengths:**

1. The finding that an SAE's interpretability score doesn't correlate well with steering efficacy is interesting.
2. The authors sweep over a comprehensive set of SAEs from different models.

**Weaknesses:**

1. The result that SAE interpretability score isn't well-correlated with steering score is interesting, but I think it would be much more valuable to present an analysis of why this is the case. Do interpretable SAE features tend to have negligible steering effects? Do they tend to cause degeneration in instruction following? I feel like this result is mainly interesting insofar as it raises follow-up questions, none of which are addressed here.
2. Similarly for the result on delta token confidence selecting for features with a high steering score: Why specifically does this work? Is it because it's selecting for features that have a non-negligible steering effect? Why doesn't this metric surface many features that make model outputs incoherent? In this case, I don't feel like the stand-alone result is very important, but it could be used as a springboard for additional analysis.
3. The result on delta token confidence selecting for high steering score is presented as an improvement on selecting on interpretability, but in fact this is a very apples-to-oranges comparison. In one case, the result is that selecting for *SAEs* with a high interpretability score doesn't result in *SAEs* with a high steering score; in the other case, the selection is being done at the level of features. An apples-to-apples comparison could be to show that *SAEs* whose average feature has a high delta token confidence have high steering scores (computed in terms of Kendall's rank coefficient). Alternatively, you could test whether selecting for interpretability at the feature level also selects for high steering score. At minimum, you should remove claims like "This result validates the superiority of our method" given that you haven't actually demonstrated that delta token confidence is superior to anything. (Unless you mean that delta token confidence is superior to the other methods in table 2, in which case that should be made clear. Also, if the comparison in table 2 is being treated as a core contribution, then the authors should (1) explain what the Arad et al. (2025) S_out metric is, given that it's very similar to the delta token confidence metric used here, (2) present some analysis of why delta token confidence is more effective than S_out, and (3) motivate why introducing metrics that select for high steering score features is important.)
4. The equation given for tau_b is actually the equation for the non-tie-corrected tau_a; it's unclear which one the authors actually used.
5. It's possible that I'm misunderstanding something here, but the descriptions in the abstract and introduction of the result in section 4.3 seems incorrect. In the abstract it says "Strikingly, after selecting features with high ∆ Token Confidence, the correlation between interpretability and utility vanishes (τb ≈ 0)". However, the actual result seems to be  that selecting for high interpretability score *SAEs* does not select for *SAEs* such that "steering gain" is large. I can't actually tell what "steering gain" is in section 4.3 (it's defined as " the percentage lift of the selected-steering score over the same SAE’s base" which is too vague for me to parse), but the same term is used elsewhere in the paper to mean "the increasing in steering score after selecting for high delta token confidence features."

**Questions:**

Why are so many of the confidence intervals in table 1 identical? E.g. the upper confidence bound for every row in the "Sparsity" section is 0.3714 despite different tau_b values.

---

> ### Author Response · Authors · 2025-11-21
> **Detailed answers and experiments regarding the first and second questions**
>
> We thank the reviewer for raising important points that improved our work!
> >1. Do interpretable SAE features tend to have negligible steering effects? Do they tend to cause degeneration in instruction following? Present an analysis of why this is the case.
>
> (1) we added a new feature-level analysis in **Appendix G**. There we join Interpretability scores with steering scores for ~9k latents across three models and plot steering overall score against interpretability (**Fig.11**). As the figure shows, high-interpretability features span the full range of steering strengths, including many strongly steering latents, so the weak global correlation does not come from interpretable features being systematically weak.
>
> (2) To test whether interpretable SAE features tend to break instruction following, we align each latent’s Interpretability score with its Instruction subscore, then bucket latents into interpretability quartiles and compare instruction-following quality across buckets.
> |Interp_quantile|Instruction score (mean) | Instruction score(median)|
> |-|-|-|
> |Q1 (low)|1.9733| 2.0|
> |Q2|1.9541|2.0|
> |Q3|1.9400|2.0|
> |Q4 (high)|1.9507|2.0|
>
> The mean Instruction score stays tightly between 1.94 and 1.97 on a 0–2 scale, and the median is 2.0 in all quartiles. This indicates that highly interpretable features are not systematically more likely to cause degeneration in instruction following.
>
> (3) Regarding '*why there is a gap*', our view is that **the root cause lies in the training method of SAE**. The core objective is to reconstruct and obtain more monosemantic basis in a higher-dimensional space. This is one of the reasons why SAE's features can be explained. When we focus on improving SAE's ability to reconstruct the input activations, we overlook SAE's steering utility. To confirm this, we performed a pairwise analysis of the two core metrics (higher is better) for evaluating the reconstruction effect after SAE training: (i) **CE loss score** and (ii) **Explained Variance**, with steering score. We have expanded our analysis in the revised version (**Appendix K**).
> | Pair|$\tau_b$|SE|95% bootstrap CI|perm $p$ (H0: $\tau_b$ = 0)|
> |-|-|-|-|-|
> | Interpretability–CE loss score|-0.433| 0.056| [-0.539, -0.322]|0.0002|
> | Interpretability–Explained variance |-0.405|0.059|[-0.521, -0.289] |0.0002|
> | CE loss score–Steering|-0.243|0.067|[-0.377, -0.112]|0.0006|
> | Explained Variance–Steering|-0.195|0.066|[-0.323, -0.062]|0.0062|
>
> As shown in the table, The reconstruction metrics show only a **weak association with steering** ($|\tau_b|$ ≈ 0.2), but a **stronger association with interpretability** ($|\tau_b$| ≈ 0.4). In other words, the training objective is much more predictive of which SAEs look good under interpretability metrics than of which SAEs are good for steering, and is approximately orthogonal to steering utility. There is a mismatch between the training objective (reconstruction) and steering utility, so we think the interpretability–utility gap is fundamental problem **in the current training paradigm, where reconstruction fidelity—not steering—is the primary optimization target.**
>
> >2. Similarly for the result...Why specifically does this work...Why doesn't this metric...make model outputs incoherent?
>
> (1) Our view is that Δ Token Confidence works because it explicitly measures something that SAE pretraining never optimizes for: how much the feature change actually moves the model’s output distribution.
>
> During SAE pretraining, the objective is purely reconstruction-centric: learn sparse features that accurately reconstruct the internal activations. These features form a basis for the representation space—essentially 0-th order information about “what is present” in the activations.
>
> Δ Token Confidence is different in kind. By construction, it looks at how the model’s token-level confidence over candidate continuations changes when we intervene on a feature. In other words, it captures the sensitivity of the output distribution to perturbations along that latent direction. **Features with high Δ Token Confidence are exactly those where a small intervention induces a large, directional shift in the logits, which is precisely what effective steering requires.**
>
> (2) In the revised version we added analysis in **Appendix H (Fig. 12)**. There we compare Instruction subscores in three cases. In all cases, the instruction score remains very close, rather than collapsing.
>
> This is because **incoherence is primarily controlled by the steering factor $\alpha$** in the intervention $x^{\text{steer}} = x + (\alpha m_f)\cdot v_f$, which AxBench explicitly tunes to balance concept shift against instruction following and fluency. Our metric doesn't push $\alpha$ higher, instead, it selects features for which even small interventions already induce strong, directional logit shifts.

---

> > ### Author Response · Authors · 2025-11-21
> > **Detailed answer and experiment regarding the third question**
> >
> > >3. An apples-to-apples comparison could be...Explain what...`S_out` metric is...present some analysis of why delta token confidence is more effective than `S_out`...why introducing metrics...is important.
> >
> > (1) We followed the reviewer’s  suggestion by first doing SAE-level analysis, with SAEs whose average feature has a high Δ token confidence.
> > |Pair|$\tau_b$|SE|95% bootstrap CI|
> > |-|-|-|-|
> > |Interpretability-Steering|0.2979|0.0661|[0.1591, 0.4191]|
> > |Δ Token Confidence-Steering |0.4136|0.0610|[0.2877, 0.5266]|
> >
> > As the table shows, the rank correlation between $\mu_{\Delta \text{TC}}(\theta)$ and $g(\theta)$ ($\tau_b \approx 0.41$) is clearly stronger than that between Interpretability Score and steering ($\tau_b \approx 0.30$). This indicates that **SAEs whose average feature has higher Δ Token Confidence tend to have higher steering scores**.
> >
> > (2) To directly address whether selecting features by interpretability also selects for high steering utility, we add a controlled feature-level comparison in **Appendix I (Fig.13)**. For each SAE, we evaluate three steering configurations: using all features (SAE Base), using only the top10% Interpretability features, and using features selected by Δ Token Confidence.
> >
> > From Fig 13, interpretability-based selection yields sometimes negative lift relative to the base (≈−35% for Gemma-2-2B, +10% for Qwen-2.5-3B, −19% for Gemma-2-9B) across models, whereas Δ Token Confidence consistently achieves large gains of roughly 1.1–1.5× over the same base. This shows that feature-level interpretability is a weak proxy for steering utility, while **our method reliably shows stronger effects.**
> >
> > (3) For each feature, `S_out` uses logit-lens procedure selects a representative token set $M$ and then observe how much the rank + probability of this set of tokens in the output distribution increases with or without this feature. This clarification has been added to the revised manuscript in **Section 4.2**.
> >
> > (4) `S_out` is completely **bound to the set of tokens $M$ represented by the logit-lens**, only considering the rank-weighted probability changes of this set of tokens. Δ token confidence, on the other hand, considers the probability distribution changes across the entire output distribution, and whether the tokens in the logit-lens set are not hard-bound to it. Therefore, it is not affected by "selecting the wrong set of representative tokens". We verify this through comparative experiments.
> >
> > | Field|Value|
> > |-|-|
> > |Model|Gemma-2-2B|
> > | Layer/Feature ID|12/7445|
> > | `S_out` (high) |0.183|
> > | Δ token confidence(low)|0.037|
> > | Steering score (overall)|0.000|
> > | Logit-lens tokens (top-10)| `['<eos>', '\n\n', '▁And', '▁The', '.', '▁This', '▁Also', '▁It', '\n', 'The']`|
> > | Concept description|`legal terminology and concepts related to court cases and appeals`|
> > |prefix|I heard that|
> > |No steer| I heard that there are no more cars coming in 2020, and I'm a little confused.|
> > |After steer| I heard that there is a new way to become a super rich, and it is very simple.|
> >
> > It can be seen that the effectiveness of `S_out` depends on the specific token group selected by logit-lens. If this group mismatches with the concept description of the feature, it will directly lead to misjudgment. In contrast, our method focuses on the overall confidence change of the top-k distribution of the model output, without binding to any preset token group. This avoids mismatch interference and tends to select those latents that **truly change the model's output behavior across various instructions**.
> >
> > (5) To answer why it is important to introduce metrics that select for high steering score features, we need to know that many studies use SAE for steering: leveraging SAE features to control model outputs, mitigating issues like hallucinations [1] and biases [2].
> >
> > - Steering ability is crucial, but researchers often don’t know the steering effectiveness of a pretrained SAE, as evaluating steering typically requires expensive and time-consuming LLM judge evaluations. The first motivation is: **Can the interpretability of SAE serve as a prior indicator of its utility?** If so, users could estimate its suitability for downstream control tasks without costly steering evaluations.
> >
> > - We find that interpretability correlates weakly with steering performance, and features selected by interpretability often have weak steering abilities. Thus, the second motivation is: **Introducing a new metric that selects for features with high steering scores**, which is essential for using SAE features in downstream tasks. We will clarify this point further in the revised version.
> >
> > [1] Do I Know This Entity? Knowledge Awareness and Hallucinations in Language Models ICLR 2025
> >
> > [2] Evaluating Feature Steering: A Case Study in Mitigating Social Biases Anthropic Blog

---

> > > ### Author Response · Authors · 2025-11-21
> > > **Detailed answers and experiments for questions four, five, and six.**
> > >
> > > >4. The equation given for `tau_b` is actually the equation for the non-tie-corrected `tau_a`
> > >
> > > We use tie-corrected Kendall $\tau_b$ implementation for all rank-correlation statistics. On our data, tie-corrected $\tau_b$ is numerically very close to the simple average-concordance version and does not affect any qualitative conclusions. In the revised version (**Section 3.2**), we (i) explicitly state that all experiments use tie-corrected $\tau_b$, and (ii) replace Eq. (3) with the full $\tau_b$ formula.
> > > >5. The actual result seems to be that selecting for high interpretability score SAEs does not select for SAEs such that "steering gain" is large. I can't actually tell what "steering gain" is in section 4.3
> > >
> > > We thank the reviewer for pointing this out and apologize for the confusion caused by our definition. The previous draft mixed two different notions (“steering score” vs. “steering gain”), which arose from a mix-up between draft versions in the submitted manuscript.
> > >
> > > In the revised version, we unify “utility” to mean the steering score itself rather than a relative gain. Table 3 now reports $\tau_b\big(\mu(\theta), g_{\text{high}}(\theta)\big)$, where $\mu(\theta)$ is the interpretability score and $g_{\text{high}}(\theta)$ is the steering score recomputed after selecting high Δ Token Confidence features for each SAE. This is directly comparable to Table 1, which reports $\tau_b\big(\mu(\theta), g_{\text{base}}(\theta)\big)$ using the original steering score $g_{\text{base}}(\theta)$.
> > >
> > > In the updated Table 3, the measured value is **$\tau_b\big(\mu(\theta), g_{\text{high}}(\theta)\big) = 0.0823$** which is numerically close to zero and much smaller than the baseline correlation in Table 1 (τ$_b$ ≈ 0.30). This is what we mean in the abstract: after restricting attention to high-utility features, interpretability no longer provides a meaningful signal for ranking SAEs by steering score. This is consistent with the core conclusion of this paper: **the interpretability–utility gap widens among high-utility features**. We have updated the abstract, introduction, and Section 4.3 accordingly in the latest revision to make this distinction clear.
> > >
> > > >6. Why are so many of the confidence intervals in table 1 identical? E.g. the upper confidence bound for every row in the "Sparsity" section is 0.3714 despite different `tau_b` values.
> > >
> > > In the original version of Table 1, the subgroup “95% CI” entries were not confidence intervals centered around the observed $\hat\tau_b$.
> > > Instead, they were **permutation-based null intervals:** for each subgroup, we fixed the interpretability scores $\mu(\theta)$, randomly permuted the steering scores $g(\theta)$ many times, recomputed Kendall’s $\tau_b$, and reported the 2.5% and 97.5% quantiles of this null distribution under $H_0!:\tau_b=0$.
> > >
> > > To avoid misunderstanding, in the revised manuscript (**Table 1 and 3**) we switch all reported ‘95% CI’ entries to bootstrap intervals around $\hat\tau_b$ and it is explained in detail in the caption. In this way, each subgroup with different `tau_b` values will have different confidence intervals, making them easier to read.

---

> > > > ### Comment · Reviewer_AA1u · 2025-11-24
> > > >
> > > > Thank you for the detailed response.
> > > >
> > > > One follow-up question: When you select on steering score, how are you choosing the selection threshold? Is it by picking the 10% of SAE features with the highest scores, like you did for interpretability selection in the new figure 13? If not, then I'd request that you align between the two selection methods to make sure that the observations here are about the underlying metric rather than about the choice of threshold.
> > > >
> > > > Pending an answer to the above, I'm planning on raising my score to 4. I don't think I can go higher because I still feel like these results have limited importance without a supporting analysis of why selecting on interpretability score/delta token confidence does/doesn't improve steering score. While I appreciate the additional experiments, none of them seem to clarify what is driving this effect.

---

> ### Author Response · Authors · 2025-11-24
> **Supplementary Experiments and Insight Explanation**
>
> We sincerely appreciate your thoughtful follow-up and your willingness to reconsider your assessment of our work！
> > Q1: How are you choosing the selection threshold?
>
> In Figure 13, interpretability-based selection uses the top 10% of features for each SAE. For Δ Token Confidence we also use a top-k scheme with the same subset size: for each SAE we rank features by Δ Token Confidence and retain the same number of features as in the interpretability condition, so any difference in steering performance comes from the metric rather than the threshold.
>
> > Q2: These results have limited importance without a supporting analysis of why selecting...does/doesn't improve steering score.
>
> **1. Empirical Analysis (cross-comparison).** Appendix G now contains a joint analysis:
> - Interp vs. Steering for every latent, and
> - The reverse view, with the top steering-quartile latents highlighted.
>
> Empirically we see:
>
> - Among high-Interp Score features, the Steering Score ranges from near 0 to the maximum value; there is no band where “high interpretability ⇒ weak steering”.
> - Among top-Steering Score features, the Interp Score again spans almost the full range; only for Qwen-2.5-3B do the highest-steering features show slightly higher interp, which explains why SAEs in this model in Appendix J shows improvement.
>
> Together, these results show that at the feature level, **interpretability and steering strength behave almost like orthogonal coordinates.**
>
> **2. Theoretical Analysis.** The SAE is trained with a reconstruction-centric loss
> $$
> \\min\_{W\_{\\text{enc}},W\_{\\text{dec}}}\\mathbb{E}\_x\\Big[\\big\\|x - W\_{\\text{dec}} \\sigma(W\_{\\text{enc}} x)\\big\\|\_2^2 + \\lambda \\|f(x)\\|\_0\\Big]
> $$
> where $x \\in \\mathbb{R}^d$ is the hidden state of the base model, and $f(x) = \\sigma(W\_{\\text{enc}} x)$ is the sparse feature latent. We denote by $p(x)$ the base model's next-token distribution when its hidden state is $x$.
>
> - The **Interpretability Score** depends on how reliably an LLM judge can infer “latent $i$ is active” from the context; it is a property of the *activation pattern* $f\_i(x)$.
>
> - The **Steering Score**, in contrast, depends on what happens when we *move the state* along $v\_i$, e.g.
>   $$
>   x^{\\text{steer}} = x + \\alpha v\_i
>   $$
>   which to first order changes a utility $U$ by
>   $$
>   \\Delta U \\approx \\alpha\\, \\nabla\_x U(p(x))^\\top v\_i.
>   $$
>   Steering strength is thus controlled by the **directional derivative** $\\nabla\_x U(p(x))^\\top v\_i$.
>
> Since the remaining network is a fixed differentiable map from $x$ to $p(x)$, any output-based utility can be written as a scalar $U(p(x))$ with sensitivity to changes in $x$ given by $\nabla_x U(p(x))$.
>
> The SAE loss never optimizes $\nabla_x U(p(x))^\top v_i$; it only cares that the $v_i$ reconstruct $x$ well. In high dimension, decoder directions that are good for reconstruction need not align with the behavior gradient $\nabla_x U(p(x))$, and the LLM judge’s ability to name the activation pattern is largely independent of this alignment. Thus high Interp Score does not predict large $|\nabla_x U(p(x))^\top v_i|$ and therefore does not reliably increase the Steering Score.
>
> Our $\\Delta$ Token Confidence metric is designed precisely to pick up this directional effect. For feature $i$,
> $$
> \\Delta C\_k(i) = \\mathbb{E}\_x\\big[C\_k(p(x + \\alpha v\_i)) - C\_k(p(x))\\big] \\approx \\alpha\\,\\mathbb{E}\_x\\big[\\nabla\_x C\_k(p(x))^\\top v\_i\\big]
> $$
> where $C\_k$ is the negative log top-$k$ confidence computed from $p(x)$. High $\\Delta$ Token Confidence selects directions with large causal impact on the logits, which is why filtering by this metric consistently raises the Steering Score, whereas filtering by Interpretability Score does not.
>
> **3. Core Insight.** Our goal is not to claim that Δ Token Confidence is a universally superior steering method; rather, we use it as a way to choose high-utility features. Our central claim is  **there is a substantial interpretability–utility gap, and this gap **widens** among the very SAEs whose features are actually effective for steering.**
>
> This gap matters for two reasons:
>
> - In practice, researchers often hope to use Interpretability Score as a **default proxy** for steering utility, because SAE is characterized by its interpretability and running evaluations with LLM judges is expensive. Our results show this proxy is unreliable.
> - Many steering papers implicitly **assume** “good interpretability ⇒ effective to steer with”. Our analysis shows that this assumption does not hold: interpretability and steering utility are complementary but only weakly coupled.
>
> The feature-selection analysis is a diagnostic step to see whether the gap shrinks or widens when select high-utility features. **Our results highlight this gap and suggest that the current SAE training should be altered and revised.** This is left as a future work for building SAEs that has a unified power for both interpretability and utility.

---

### Author Response · Authors · 2025-12-02
**Rebuttal Summary**

Dear Reviewers, ACs, SACs, and PCs,

We sincerely thank you and all reviewers for your time and thoughtful feedback on our submission. We are encouraged that our work is viewed as **providing surprising evidence that interpretability scores are not reliable indicators of utility**, challenging a common assumption in SAE evaluation (AA1u, MCfE, 2ZdN, vNrz). It is further recognized for its **large-scale, comprehensive study of 90 SAEs and hierarchical analysis across models and architectures** (MCfE, vNrz). This framing **could inspire new evaluation metrics and SAE designs** in the community (2ZdN, MCfE)! During the rebuttal, we conducted a series of analyses, incorporated them into the revised manuscript  (mainly Appendices G–K). Next, we briefly summarize the reviewers' concerns and explain how our responses **successfully** address these concerns.

---
### Summary of Rebuttal

The initial review scores are **8, 6, 2, and 2**. **Reviewer AA1u** mainly focused on **(i)** the deep reason behind the interpretability–utility gap and on **(ii)** whether our metric genuinely improves feature selection. **Reviewer MCfE** mainly questioned **(iii)** how we define interpretability and **(iv)** whether our statistical testing is strong enough to support claims. The two positive reviewers **(2ZdN, vNrz)** mainly requested **(v)** broader validation (more steering baselines, more interpretability metrics, and qualitative examples). Specifically:

- We ran joint feature-level empirical analyses and added reconstruction theoretical analysis (Appendices G–H, K) to deeply explain the interpretability–utility gap.

- We performed SAE-level and feature-level selection analyses (Appendix I), directly comparing Δ Token Confidence against interpretability-based selection and the output-based S_out baseline under matched subset sizes.

- We additionally evaluated four interpretability metrics, and the results are **fully consistent with the core insight of our paper**, which directly addresses the reviewer **MCfE**’s primary concern: *“Sure, I appreciate that...there is general use of this metric. Thanks for doing the additional experiments with the other metrics.”*

- We systematically added bootstrap confidence intervals and hypothesis tests (Appendix J), and updated Tables 1 and 3 according to the reviewer’s suggestions, which directly addresses another central concern of the reviewer **MCfE**: *“This analysis is clear ... thanks for doing this.”*
- We conducted additional analyses using correlational interpretability metrics, several non-SAE steering baselines, and qualitative max-activating examples, which **further strengthens our conclusions across a broader range of metrics and baselines.**

Importantly, the two critical reviewers **AA1u** and **MCfE** responded positively to these changes. Reviewer **AA1u** wrote: *“Pending an answer to the above, I'm planning on raising my score to 4.”* and Reviewer **MCfE** commented: *“Thanks for doing the additional experiments. This is really interesting to see! … That makes much more sense now.”*   **While their numeric scores have not yet been updated in the system, these comments indicate that our rebuttal successfully addressed the main concerns from both critical reviewers.** The core concern of Reviewer **2ZdN** about the notion of interpretability is aligned with **MCfE’** s first concern and is directly addressed (see point 3 above), and the additional experiments and examples requested by **2ZdN** and **vNrz** have all been incorporated in the revised manuscript.

### Addressing Follow-up Requests

After the initial rebuttal, once their original concerns had been addressed, Reviewers **AA1u** and **MCfE** raised **more exploratory follow-up questions about the underlying reasons**. In response, we proposed our view and supported it with empirical analyses and theoretical proof.

- **Follow-up from Reviewer AA1u**
  We performed empirical analysis and theoretical proof showing that interpretability and steering strength behave almost like orthogonal axes, so filtering by Interpretability alone does not reliably raise steering, while Δ Token Confidence directly targets directions with strong causal impact on the logits and therefore consistently improves steering.
- **Follow-up from Reviewer MCfE**
  We provided empirical explanation that an inverted-U analysis over intervention strength and steering score explains the resulting negative Kendall $\tau_b$ and clarifies why the two metrics capture fundamentally different properties.

---

Taken together, based on their post-rebuttal comments, we believe the two initially negative reviewers **(AA1u, MCfE)** are now broadly satisfied with our responses and additional experiments, and are likely to increase their evaluation during the rebuttal.

We are deeply grateful to the ACs for their extra time, effort, and thoughtful consideration in  handling our submission. Thank you once again for your careful evaluation!

Authors

---

### Meta-Review · Area_Chair_jJ56 · 2025-12-03

**Summary:**

The initial recommendation regarding this paper was primarily informed by concerns about the depth of causal analysis and the methodological fairness of the comparisons. Reviewers emphasized that while the identification of an "interpretability–utility gap" is empirically interesting, the work did not sufficiently explain why this gap exists or whether interpretable features cause model degeneration or simply lack causal influence on the output. Furthermore, the operationalization of "interpretability" was criticized as too narrow for excluding interventional metrics, which are crucial for a complete assessment. Finally, the comparison between SAE-level interpretability selection and the proposed feature-level Δ Token Confidence was flagged as an unfair "apples-to-oranges" evaluation. Despite improvements in the rebuttal, concerns regarding statistical rigor and the fundamental significance of the findings persisted.

However, I have a few key points beyond this summary that I want to raise. This paper required a substantial amount of time for me to reach a conclusion. First, my main criticism, which may have confused reviewers and led to questions about "operationalizing" utility and interpretability and the relation to steering, is the following. (1) The paper should make it crystal clear that its conclusions are not about interpretability in general but about interpretability using SAE features. This is a major distinction that the authors failed to clarify. In fact, the problem begins with the title of the paper. The perceived claim is much larger than what is actually shown. In recent years, SAEs have not delivered on many of their promises, and simple linear probing in some cases matches or outperforms SAE features. (2) Utility, as measured here through steering, is not the full notion of utility that interpretability aims to address. I agree with reviewers that the current framing can be misleading. Instead of referring broadly to utility, the authors need to clearly specify the sense in which steering is being used and be explicit about this limitation. (3) Regarding the explanation for why their proposed token-confidence metric works better: although I appreciate the connection to first-order Taylor approximation and directional derivatives, the argument remains somewhat weak. Nevertheless, the reasoning is useful and should be included in the appendix.

One reviewer raised their score after rebuttal clarifications, and another had an overall positive impression of the paper post-rebuttal. Overall, I believe the paper provides value to the community, particularly the analysis connecting SAE features to steering, though less so the proposed token-confidence approach. The remaining concerns, in my judgment, do not warrant rejecting the paper, and given its contribution, I recommend accepting it.

However, I strongly urge the authors to take my comments seriously and reflect them in the camera-ready version.

**Reviewer Concerns:**

see above.

**Reviewer Scores:**

see above.

---

### Decision · Program_Chairs · 2026-01-26

Accept (Poster)